# Linear and nonlinear correlation estimators unveil undescribed taxa interactions in microbiome data

Huang Lin [1], Merete Eggesbø [2] & Shyamal Das Peddada [1] ✉

It is well-known that human gut microbiota form an ecosystem where microbes interact with each other. Due to complex underlying interactions, some microbes may correlate nonlinearly. There are no measures in the microbiome literature we know of that quantify these nonlinear relationships. Here, we develop a methodology called Sparse Estimation of Correlations among Microbiomes (SECOM) for estimating linear and nonlinear relationships among microbes while maintaining the sparsity. SECOM accounts for both sample and taxon-specific biases in its model. Its statistical properties are evaluated analytically and by comprehensive simulation studies. We test SECOM in two real data sets, namely, forehead and palm microbiome data from college-age adults, and Norwegian infant gut microbiome data. Given that forehead and palm are related to skin, as desired, SECOM discovers each genus to be highly correlated between the two sites, but that is not the case with any of the competing methods. It is well-known that infant gut evolves as the child grows. Using SECOM, for the first time in the literature, we characterize temporal changes in correlations among bacterial families during a baby's first year after birth.

In any given ecosystem, such as microbial ecology in the gut, members maintain complex interactions among themselves for the growth and stability of the ecological community. Some bacteria display mutualism where they help or benefit from each other, commensalism where one benefits while the others are not affected, parasitism or mutual exclusion where one may benefit at the expense of another[1]. One needs to understand perturbations or changes in the interactions among microbiota within and between ecosystem(s) because they can potentially impact various health outcomes, such as obesity[2], inflammatory bowel diseases[3], HIV[4], and so on. To make such assessments, there is an urgent need for statistical and computational methods to characterize interactions among microbiota.

Although there exists considerable literature on methods for performing differential abundance analysis and analysis of diversity[5,6], methods for describing interactions among microbiota are not as well developed. As the first step to characterize interactions, one typically is interested in describing the correlation coefficient between a pair of microbes.

16S ribosomal RNA (rRNA) and other high-throughput sequencing techniques enable the profiling of microbial communities, revealing the abundances and phylogeny of microbial populations across diverse ecosystems. The observed counts of taxa are constrained by an arbitrary sequencing depth (library size), which is due to a fixed upper bound on the number of reads per sequencing instrument. Thus, the observed counts are an unknown fraction of the underlying true abundance of taxa in a unit volume of the ecosystem[6,7] such as 1 g biomass of the gut. Consequently, they are relative quantities and hence compositional data[7–13]. It is well-recognized in the literature that due to the compositionality of the observed microbiome data, standard methods such as Pearson or Spearman correlation coefficient for these data are theoretically invalid and result in misinterpretation of the data[9,14–17]. The effect of compositionality on these methods is

[1]Biostatistics and Bioinformatics Branch, Eunice Shriver Kennedy NICHD, NIH, Bethesda, MD, USA. [2]Norwegian Institute of Public Health, Oslo, Norway.
✉ e-mail: shyamal.peddada@nih.gov

particularly pronounced when the number of taxa is small, such as microbiome data evaluated at the phylum level, and when the microbial diversity is small[14]. Additionally, sequencing efficiencies are not necessarily the same for each taxon, which leads to a taxon-specific bias when some taxa are preferentially measured over others during the sequencing experimental workflow[18]. A concrete example of this bias can be seen by the differential sequencing efficiencies between gram-negative and gram-positive bacteria. Gram-positive bacteria have strong cell walls, which makes them harder to extract than gram-

negative bacteria in the data preparation step. Thus, gram-positive bacteria may be underrepresented in the observed abundances. We illustrate the problems of using standard concepts of correlation coefficients with the help of a toy example shown in Fig. 1. A total of 100 taxa were generated from a log-normal distribution. For simplicity of visualization, scatter plots of log abundances corresponding to 5 taxa are shown in the figure. Data were generated so that the underlying true log abundances for taxa T1 and T3 are perfectly linearly correlated, whereas taxa T2 and T4 are parabolic in their relationship,

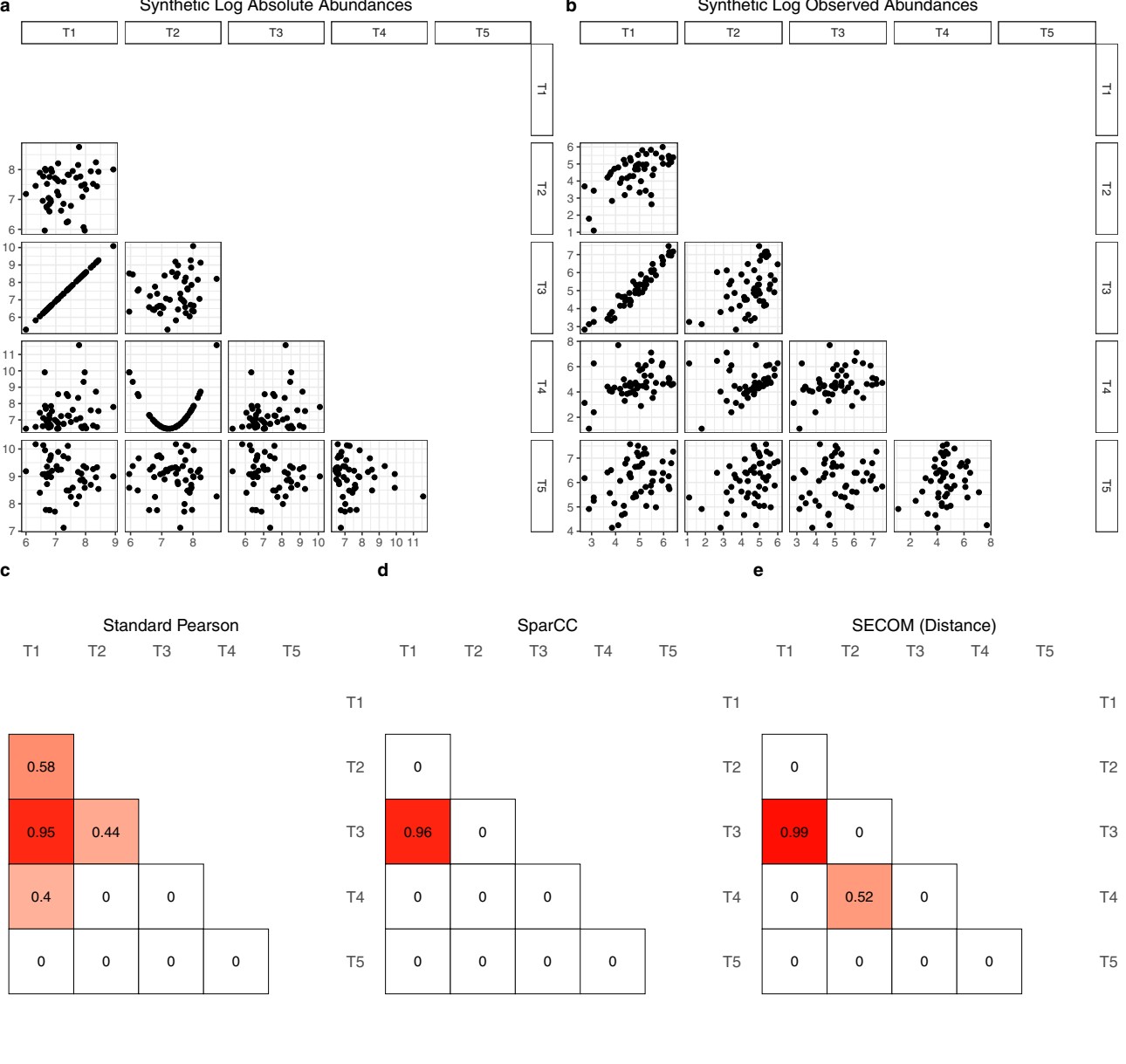

**Fig. 1 | A toy example showing the differences between correlation methods.** All scatter plots and correlations provided in the figure are based on log abundances. **a** A total of 100 taxa were generated, where the true abundances in the ecosystem were generated using log-normal distributions. Taxa T1 and T3 are perfectly linearly correlated on a log scale, and taxa T2 and T4 are quadratically related. The remaining taxa are uncorrelated. Only the first 5 taxa (T1 to T5) are provided for the simplicity of exposition. Pairwise scatter plots of log abundances are provided in the lower triangle of the matrix. **b** Observed abundances were generated by incorporating sample-specific sampling fractions and taxon-specific sequencing efficiencies to the true abundances. Pairwise scatter plots of log abundances are provided in the lower triangle of the matrix. **c**–**e** Pairwise correlation coefficients using different methods. **c** The standard Pearson correlation coefficient (two-sided $p$ value < 0.005, based on t-distribution, not corrected for multiple comparisons), **d** SparCC (estimate > 0.3), and **e** SECOM using distance correlation measure (two-sided $p$ value < 0.005, based on the permutation test[34], not corrected for multiple comparisons). The correlation coefficient ranges from −1 to 1, color-coded by blue to red, correspondingly.

and the remaining pairs are uncorrelated (Fig. 1a). These correlations are completely distorted in the observed counts (Fig. 1b) due to differential sampling fractions[6,7] (across samples) and differential sequencing efficiencies (across taxa). Without properly addressing these two sources of biases results in erroneously large standard Pearson correlation coefficients with significant *p* values (Fig. 1c). Due to variations in sampling fractions and sequencing efficiencies, the distribution of *p* values based on standard Pearson correlation coefficient is far from a uniform distribution, represented by the diagonal line (Supplementary Fig. 1).

A second challenge is that the next-generation sequencing (NGS) technologies yield high-dimensional data, where the number of microbiota *d* often far exceeds the sample size *n*. Consequently, the estimated correlation matrix of the microbiota is singular (positive semi-definite matrix)[19] unless a condition of sparsity is imposed on the population correlation matrix. In simple terms, sparsity means that many pairwise correlations are zero. Thus, compositionality aside, due to the high dimensionality, the sample Pearson correlation matrix results in a biased and inconsistent estimator of the correlation matrix[20]. A host of approaches are available to estimate the covariance matrices in high dimensions[21–29]. However, since the observed high-dimensional microbiome data contain a large number of zeros and are compositional, the problem of estimating the correlation matrices in the present setting is challenging. It is well-known that microbiome abundance tables may contain as many as 90% zero entries[6,30], therefore it is reasonable to assume that most taxa are not correlated or weakly correlated with each other. Thus, the sparsity assumption is reasonable in the present context.

Thirdly, since microbiota have complex inter-dependencies, some relationships may be nonlinear. Consider, for example, the relationship between *Ruminococcaceae* and *Enterobacteriaceae* of 4-month-old infants (Supplementary Fig. 2). The data were obtained from the Norwegian Microbiome study (NoMIC)[31,32]. While the linear fit seems reasonable (adjusted $R^2 = 0.53$), a fourth-degree polynomial appears to fit the data better (adjusted $R^2 = 0.84$). In more complex settings, nonlinear relationships among taxa are ubiquitous[33]. The Pearson correlation coefficient is designed for quantifying linear relationships and not for quantifying nonlinear relationships. Recent methods such as Sparse Correlations for Compositional data (SparCC)[14], proportionality[15], and Sparse Inverse Covariance Estimation for Ecological Association Inference (SPIEC-EASI)[16] were also developed for quantifying linear relationships and may not be suitable for nonlinear relationships. For instance, the SparCC correlation between T2 and T4 in Fig. 1a is zero (Fig. 1d), suggesting no linear relationship. Researchers often misconstrue that two taxa are independent if the Pearson correlation coefficient between them is zero, which is not correct.

To quantify nonlinear correlations between a pair of variables, Szekely et al.[34] introduced the concept of distance correlation for the Euclidean space data. The difference between distance correlation and Pearson correlation coefficient is illustrated in Supplementary Fig. 3. Both measures provided a value of 1 when *x* and *y* are perfectly linearly correlated. However, when $y = (x-5)^2$, the Pearson correlation coefficient is 0, whereas the distance correlation coefficient is ~0.5. It is important to note that a value of zero distance correlation implies statistical independence between two variables, whereas a zero Pearson or SparCC correlation coefficient only implies a lack of linear relationship.

In this work, we introduce a methodology called Sparse Estimation of Correlations among Microbiomes (SECOM), which provides two measures of correlations between a pair of taxa, one for linear relationships and the other for nonlinear relationships using the concept of distance correlations. As can be seen in Fig. 1e, SECOM is useful for quantifying linear as well as nonlinear relationships. In the Methods section of this paper, we provide guidance on how to use these

measures when describing dependencies between a pair of taxa. The proposed methodology is statistically rigorous and accounts for compositionality and differential sequencing efficiencies by correcting for the library-specific sampling fraction[6,7] and taxon-specific sequencing efficiency[35,36]. The resulting observed counts are called "bias-corrected" counts, which refines the ANCOM-BC model[6,7]. Furthermore, as can be seen from the *p* value distribution provided in Supplementary Fig. 1, unlike the standard Pearson correlation coefficient, SECOM does not suffer from inflated false correlations between taxa. Lastly, as demonstrated in the Results section of the paper, except SECOM, none of the existing methods, such as SparCC are suitable for correlating data from multiple ecosystems. The performance of SECOM is evaluated using a variety of simulation studies. In addition to results based on synthetic data, SECOM is illustrated using "forehead" and "palm" microbiome data obtained in Flores et al. [37] and the infant gut microbiome data from the Norwegian Microbiome study (NoMIC)[31,32]. The statistical details are provided in the Methods section, and theoretical proofs are deferred to the Supplementary Methods.

## Results

### Performance of SECOM using simulated data

Using simulated data, in this section we compare the performance of SECOM with some existing methods, namely, proportionality[15], SparCC[14], SPIEC-EASI[16] using either the neighborhood selection (the MB method) or covariance selection (the glasso method), and standard Pearson correlation coefficient. For estimating linear correlations, SECOM uses the Pearson correlation coefficient after correcting for sample and taxon-specific biases. The thresholding approach for dealing with sparsity is denoted as "SECOM (Pearson1)", and the *p* value filtering approach, i.e., correlations with *p* values exceeding a pre-specified cutoff $\alpha$ will be set to 0s, is denoted as "SECOM (Pearson2)". For estimating nonlinear correlations (which includes linear correlation as a special case), SECOM uses the distance correlation measure with *p* value filtering approach to deal with sparsity (denote it as "SECOM (Distance)"). Discussions on how to choose a correlation measure and an approach for sparsity are provided in the Methods section. All details regarding the simulation study design are provided in the Supplementary Methods.

We begin by comparing CPU run times of various methods for computing correlation matrices using a simulated data set consisting of 200 taxa and 100 samples. All comparisons are made using RStudio with 1 CPU, x86_64-apple-darwin17.0 (64-bit), and macOS Big Sur/Monterey 10.16. Results are summarized in Supplementary Table 1. On average, SPIEC-EASI methods took nearly a hundred times more CPU time than other approaches, while SECOM (using the Pearson correlation measure in this benchmark) was competitive with other compositional methods, such as SparCC and Proportionality.

Next, we compare the performance of SECOM with other methods for estimating the correlation matrix. As a measure of accuracy, we use the following average relative norm loss as the criterion for evaluating the performance of a method:

$$\mathbb{E}\left\{\frac{\|\hat{R} - R^0\|_*}{\|R^0\|_*}\right\}.$$

We use the two commonly used matrix norms $\|\|_*$, namely, Frobenius norm ($\|A\|_F = (\sum_{ij} a_{ij}^2)^{1/2}$) and spectral norm ($\|A\|_\infty = \lambda_{max}(A)$, which is the largest singular value of *A*). The smaller the average relative norm loss, the more accurate the method is for estimating the correlation matrix. Additionally, we also evaluated the performance of various methods in their ability to discover true nonzero correlations, defined as the true positive rate (TPR), and falsely declaring nonzero correlations when the correlations are truly zero, defined as the false positive rate (FPR). Thus, the estimated TPR and FPR are as defined

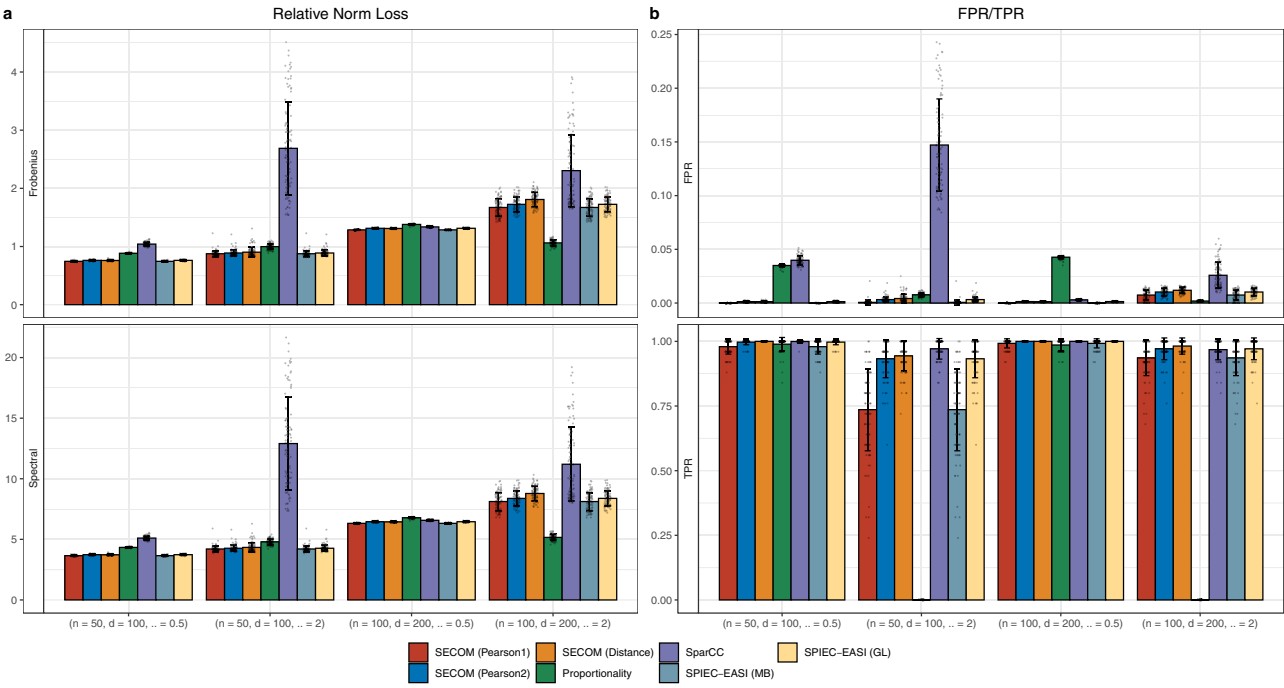

**Fig. 2 | Comparisons of estimation accuracy and false/true positive rate (FPR/TPR) for identifying linear relationships within an ecosystem.** The average relative norm loss (Frobenius/Spectral) and FPR/TPR of various correlation methods are shown in **a**, **b**, respectively. Synthetic data were generated from negative-binomial (NB) distributions. The X-axis denotes the simulation settings, which are a combination of sample size $n$, number of taxa $d$, and the dispersion parameter $\alpha$. Results are represented by the average of the corresponding measures (Frobenius/ Spectral norm loss, or FPR/TPR) ± standard errors (shown as error bars) across 100 simulation runs for each $n/d/\alpha$ setting. Data points are added to the bar charts using dots with jittering. Color and the name of the corresponding correlation methods are shown at the bottom of the graph. The results demonstrate that SECOM and SPIEC-EASI outperformed all existing methods not only in terms of estimation accuracy but also in terms of uniformly small FPR and comparable TPR.

below:

$$\text{TPR} = \frac{\#\{(l, m) : \hat{\rho}_{lm} \neq 0, \rho_{lm}^0 \neq 0\}}{\#\{(l, m) : \rho_{lm}^0 \neq 0\}},$$

$$\text{FPR} = \frac{\#\{(l, m) : \hat{\rho}_{lm} \neq 0, \rho_{lm}^0 = 0\}}{\#\{(l, m) : \rho_{lm}^0 = 0\}},$$

where $R^0 = [\rho_{lm}^0]_{l,m=1,\dots,d}, \hat{R} = [\hat{\rho}_{lm}]_{l,m=1,\dots,d}$.

Simulation studies are based on three different scenarios, namely, (1) an ecosystem where some taxa are linearly correlated and the rest uncorrelated, (2) an ecosystem where some taxa are nonlinearly correlated and the rest are uncorrelated, and (3) two ecosystems where some taxa are linearly correlated between ecosystems and the rest are uncorrelated both within and between ecosystems. Since the standard Pearson correlation coefficient performed uniformly much worse than other methods, we decided to not include it in the figures provided in the main text simply for the sake of visualization. We provided the full comparisons, including the standard Pearson correlation coefficient in the corresponding supplementary figures.

**Linear correlations.** Figure 2 summarizes the results using different approaches to estimate the correlation matrix under different sample size/taxon number/dispersion combinations ($n/d/\alpha$). As expected, under both norms, all compositional methods had uniformly smaller average relative norm loss than the standard Pearson correlation coefficient (Supplementary Fig. 4a). For example, with 100 samples, 200 taxa, and the dispersion parameter equals to 0.5, the standard Pearson correlation coefficient had approximately nine times higher average loss under both norms in comparison to SECOM (Frobenius: 9 vs. 1; Spectral: 45 vs. 5). SparCC had the second largest average loss especially when there is greater variability in the underlying true

abundances (overdispersion parameter $\alpha = 2$, Fig. 2a). SECOM methods, regardless of correlation measures and approaches for sparsity, together with the proportionality method and both SPIEC-EASI methods outperformed other methods in terms of average loss (Fig. 2a). In addition to the estimation accuracy, SECOM and SPIEC-EASI also performed well in identifying true nonzero correlations (Fig. 2b). They not only outperformed other methods substantially in controlling FPR, but they competed well in terms of TPR. However, as mentioned above, SPIEC-EASI is substantially more computationally intensive than SECOM. Under this simulation setting (100 iterations for each $n/d/\alpha$ combination), SPIEC-EASI required a CPU time of 76.15 h in comparison to 1.2 CPU hours for SECOM. Both SparCC and the standard Pearson correlation coefficient achieved high TPR in this simulation study but were subject to very high FPR as compared to SECOM and SPIEC-EASI. Note that the FPR of standard Pearson correlation can be as large as 80% (Supplementary Fig. 4b), which could make most estimates meaningless and even misleading. Although the proportionality method performed well in terms of FPR, it had very small TPR when there are larger variations in taxa abundance. For example, when the underlying true dispersion parameter increased to 2, the TPR of the proportionality method dramatically reduced to almost 0. It is also worth noting that for a fixed dispersion parameter, as the sample size $n$ and the number of taxa $d$ increased, there is an increase in TPR together with a decrease in FPR for SECOM methods. This is consistent with the concentration results described in Theorem 1. Additionally, as seen in Fig. 2b, SECOM (Pearson1) has generally smaller TPR as well as smaller FPR than the $p$ value filtering methods (SECOM (Pearson2) and SECOM (Distance)).

**Nonlinear correlations.** We compared the performance of various methods in identifying and quantifying nonlinear relationships using simulations. The settings were similar to the simulation study

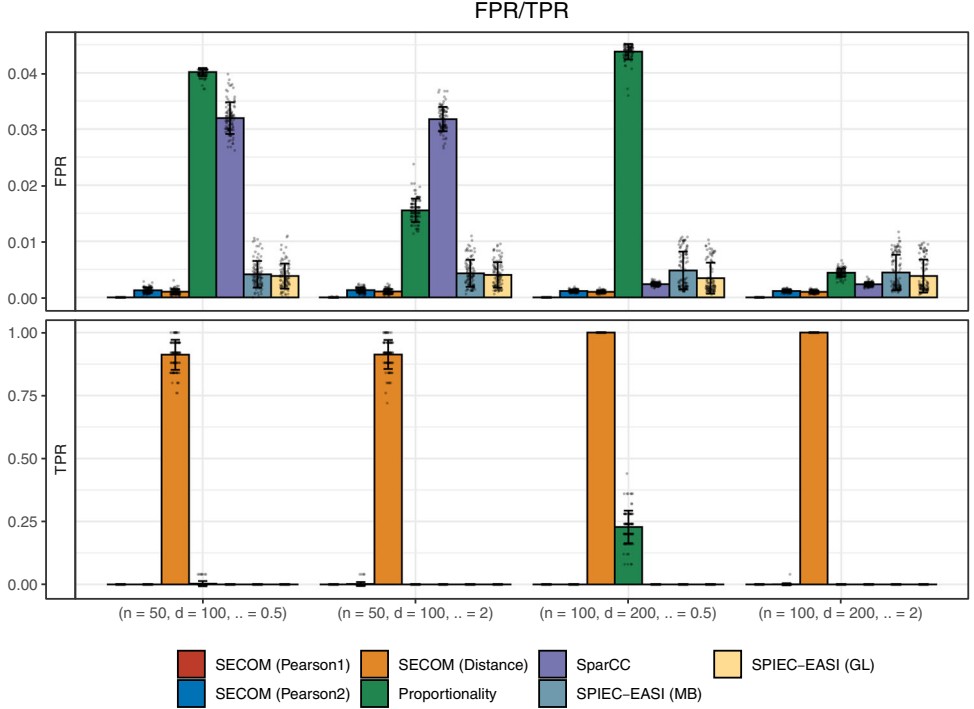

**Fig. 3 | Comparisons of false/true positive rate (FPR/TPR) for identifying non-linear relationships within an ecosystem.** Synthetic data were generated from log-normal distributions. The X-axis denotes the simulation settings, which are a combination of sample size $n$, number of taxa $d$, and the dispersion parameter $\alpha$. Results are represented by the average of corresponding measure (FPR/TPR) ± standard errors (shown as error bars) across 100 simulation runs for each $n/d/\alpha$ setting. Data points are added to the bar charts using dots with jittering. Color and the name of the corresponding correlation methods are shown at the bottom of the graph. Results showed that only SECOM controlled the FPR while maintaining high TPR. Other than SECOM, none of the existing methods identify nonlinear relationships.

conducted for linear correlations, except that each taxon was quadratically correlated with its adjacent taxon among the first 50 taxa. Since, with the exception of SECOM, none of the methods are designed to measure nonlinear relationships, we only compared these methods in their ability to identify true nonzero correlations. For estimating nonlinear correlations, SECOM implements the $p$ value filtering approach for sparsity but not the data-driven thresholding approach because it involves splitting data into training and test sets, which requires a much larger sample size when quantifying nonlinear relationships. The sparsity strategies for SparCC and the proportionality method and the tuning parameters for SPIEC-EASI used in this simulation set-up are the same as those used in the linear case mentioned earlier. As expected, all competing methods, except the standard Pearson correlation coefficient, resulted in zero correlations for nonlinear relationships (Fig. 3 and Supplementary Fig. 5) with TPR ≈ 0. Although the standard Pearson correlation coefficient achieved a TPR ≈ 50%, its FPR was sometimes even higher, leading to spurious correlations (Supplementary Fig. 5). On the other hand, SECOM (Distance) successfully achieved an almost perfect TPR (100%), and also had substantially smaller FPR than existing methods. Note that the linear correlation methods SECOM (Pearson1) and SECOM (Pearson2) had smaller FPR than all other linear correlation methods.

**Concordance between SECOM (Pearson2) and SECOM (Distance) for linear and nonlinear relationships.** Since a linear relationship is a special case of a nonlinear relationship, hence theoretically, whenever SECOM (Pearson2) is nonzero, we would expect SECOM (Distance) to be nonzero. Similarly, whenever SECOM (Distance) is zero, we would expect SECOM (Pearson2) to be zero as well. We performed a simulation study to evaluate the concordance between SECOM (Pearson2) and SECOM (Distance). For brevity, we summarize the results for $n = 50$, $d = 100$, and $\alpha = 0.5$ when 25 pairs of taxa are linearly correlated (or nonlinearly correlated) while the rest of 4925 pairs are

uncorrelated. Results are averaged over 100 simulation runs. As can be seen from Supplementary Table 2, in the linear case, there is a very high concordance between SECOM (Pearson2) and SECOM (Distance) and very little disagreement. On average, both SECOM (Pearson2) and SECOM (Distance) identified 28 pairs of taxa to be correlated, which consisted of 25 true positives (TPs) and 3 false positives (FPs). On the other hand, both methods uniquely identified four correlated pairs, which were FPs. This suggests that when the relationships are linear, SECOM (Distance) can be as effective as SECOM (Pearson2) in estimating the correlation coefficient, although SECOM (Pearson2) can provide the direction of a relationship. In the nonlinear case, reported in Supplementary Table 3, we notice that around 26 pairs (23 TPs and 3 FPS) on average, were uniquely identified by SECOM (Distance) as correlated, while only four pairs were uniquely declared correlated by SECOM (Pearson2) but not by SECOM (Distance), and were spurious correlations. There were only two pairs of taxa on average that were found to be correlated by both methods. Thus, as expected, when the data are nonlinearly related, SECOM (Distance) appears to quantify the nonlinear relationship better than SECOM (Pearson2). These results are consistent with the results summarized in Fig. 3. See also Remark 3. To eliminate the effect of differences due to the sparsity approach from this concordance analysis, we did not include SECOM (Pearson1) but were limited to SECOM (Pearson2) and SECOM (Distance), although we do not expect major differences from these results.

**Multiple ecosystems.** Correlating taxa abundance in two or more body sites or correlating at two different time points within a site is a common problem of interest. Since the sampling fractions are not expected to be the same across ecosystems, by simple algebra, it can be shown that the correlation coefficient estimates derived from the SparCC methodology can potentially be biased. Consequently, accurate estimates of correlation coefficients may not be derived using SparCC. This phenomenon holds true for other methods such as

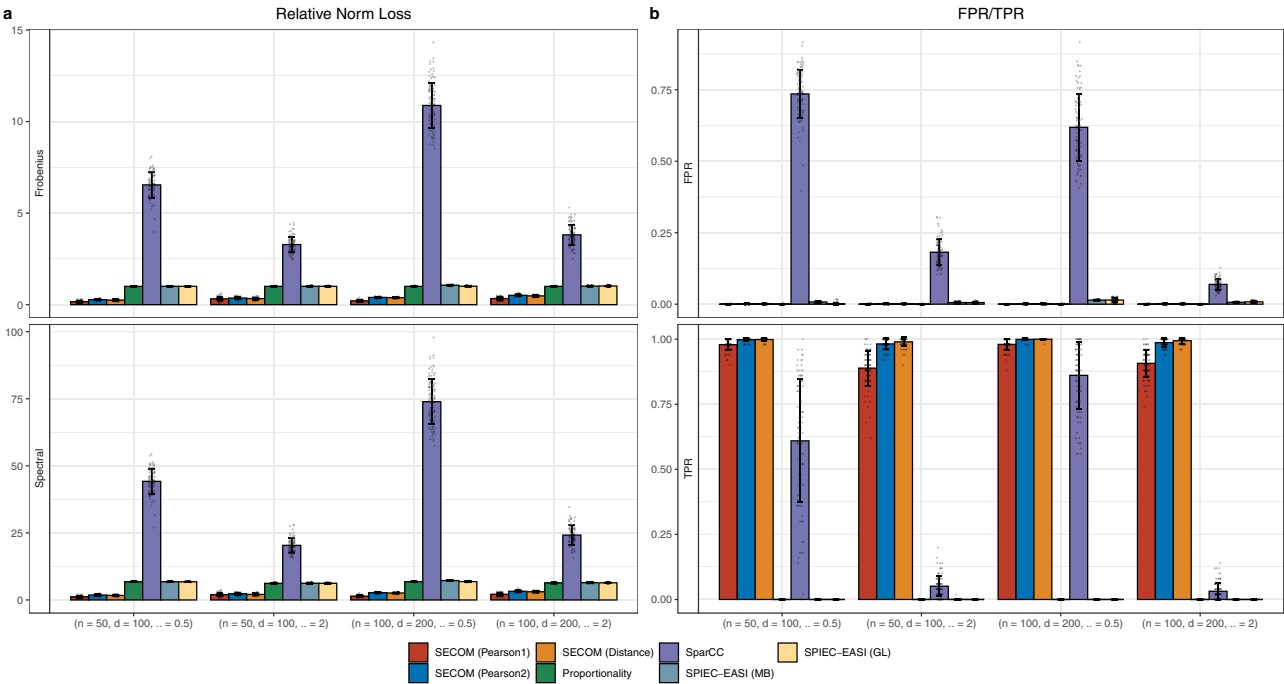

**Fig. 4 | Comparisons of estimation accuracy and false/true positive rate (FPR/TPR) for identifying linear relationships between two ecosystems.** The relative norm loss (Frobenius/Spectral) and FPR/TPR of various correlation methods are shown in **a**, **b**, respectively. Synthetic data were generated from the negative-binomial (NB) distribution. The X-axis denotes the simulation settings, which are a combination of the sample size $n$, the number of taxa $d$, and the dispersion parameter $\alpha$. Observed abundances from different ecosystems have different sampling fractions and sequencing efficiencies in each simulation run. Results are represented by the average of the corresponding measure (Frobenius/Spectral norm loss, or FPR/TPR) ± standard errors (shown as error bars) across 100 simulation runs for each $n/d/\alpha$ setting. Data points are added to the bar charts using dots with jittering. Color and the name of the corresponding correlation methods are shown at the bottom of the graph. Results demonstrate that SECOM methods are the only methods to successfully quantify microbial correlations between ecosystems.

proportionality as well as SPIEC-EASI. However, SECOM does not suffer from this problem because the method is applied to bias-corrected count data. This is amplified in the simulation study described in this section, and the "forehead" and "palm" data analyzed in the Illustration section provided below. A simulation study was conducted by generating count tables from two ecosystems with different sampling fractions and sequencing efficiencies in each simulation run. The first 50 taxa in the first ecosystem were linearly correlated with the corresponding taxa in the second ecosystem. The rest of the taxa are uncorrelated within and between ecosystems. Details of the study design are provided in the Supplementary Methods. As seen in Fig. 4 and Supplementary Fig. 6, SECOM methods outperformed all competitors in terms of both estimation accuracy (the smallest average relative Frobenius/Spectral norm loss) and FPR/TPR (the smallest FPR and the highest TPR). In general, SECOM (Pearson1) had the most precise correlation estimate, the smallest FPR, and TPR comparable to SECOM (Pearson2) and SECOM (Distance). SECOM methods achieved almost perfect TPR (~100%), with SECOM (Distance) being the best. On the contrary, standard Pearson correlation coefficient, and the existing compositional techniques, either suffered from loss of TPR or inflated FPR, making them unsuitable for detecting microbial inter-relationships across ecosystems.

In summary, SECOM performed consistently well regardless of whether the relationships are linear or nonlinear. The thresholding version of SECOM is entirely data-driven and is theoretically proved to be a consistent estimator of the true correlation matrix in the Frobenius norm (Theorem 3 in the Methods section). However, since the thresholding approach requires splitting data into training and test sets, it generally needs a larger sample size, especially when it comes to the detection of nonlinear relationships. On the other hand, the filtering version of SECOM screens the estimates of correlation coefficients if the $p$ value exceeds a pre-specified value. The filtering

approach requires a smaller sample size than the thresholding approach as there is no data splitting step. Note that the $p$ value filtering should be viewed as an approach to achieve sparsity rather than formal hypothesis testing. For simplicity of exposition, we only show the filtering approach in the following Illustration sections, while the thresholding approach gave similar results as $p$ value filtering when detecting linear relationships (data not shown).

## Illustration: Two ecosystems—forehead and palm microbiome data

Since both forehead and palm are skin based, it is intuitive to expect the two sites to share some common microbes that are highly correlated. We evaluated various methods in their ability to identify genera that are correlated using the data obtained in Flores et al.[37] on 89 subjects. For illustration purpose, we limit our analysis to the data obtained at the first visit (baseline), and we restricted it to those who did not use antibiotics. The demographic information regarding the samples is summarized in Supplementary Table 4. We selected the top five most abundant genera that are common between forehead and palm for illustration. As seen in Fig. 5a, according to SECOM, the same genera from the two sites are highly correlated. Furthermore, correlations within the site appear to be unchanged whether one computes correlations using the concatenated data from the two sites (Fig. 5a) or computes correlations using individual data from the two sites (Fig. 5b, c). However, this is not the case with other methods, as seen in Supplementary Figs. 7, 8, and 9. Firstly, to our surprise, according to these methods, none of the genera are correlated between the two skin sites, namely, the forehead and palm. Furthermore, the proportionality method (Supplementary Fig. 7) and SPIEC-EASI (using the MB method, Supplementary Fig. 9) found all genera to be uncorrelated (or nearly uncorrelated in the case of SPIEC-EASI) even within each site. The correlation coefficient estimates obtained

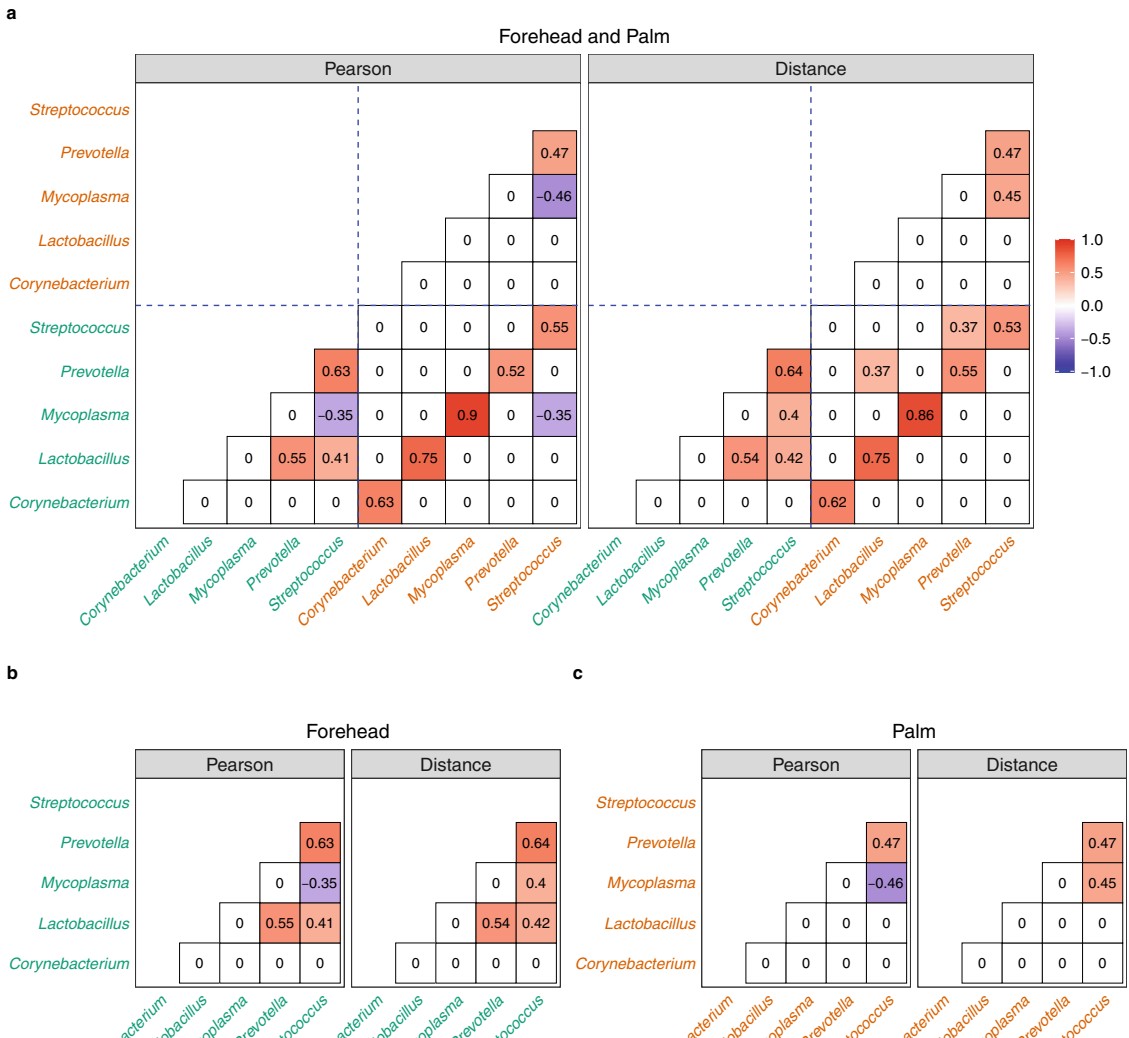

**Fig. 5 | Correlations between forehead and palm genera[37] using SECOM.**
**a** Correlation matrices calculated by concatenating forehead and palm data,
**b** Correlation matrices calculated using the forehead data alone, **c** The correlation
matrices calculated using the palm data alone. The correlation measures (Pearson
and Distance) are represented by columns. The top five most abundant genera were
selected for visualization purposes. Genera from the forehead are colored in green
and genera from the palm are colored in brown in the x and y axes. Blue dashed
lines in **a** are used to separate genera between forehead and palm. The correlation
coefficient ranges from −1 to 1, color-coded by blue to red, respectively.

by SparCC within each site changed when one computes correlations
using the concatenated data from the two sites (Supplementary
Fig. 8a) compared to estimates using individual data from the two
sites (Supplementary Fig. 8b, c). Note that the correlation estimates
based on SECOM (Pearson2) and SECOM (Distance) are very similar
whenever SECOM (Pearson2) ≠ 0, which is not surprising in view of
Remark 3 in the Methods section and the results in Supplementary
Table 2 described above.

**Illustration: Norwegian infant gut microbiome data**
We illustrate SECOM using the data obtained from the Norwegian
Microbiome (NoMIC)[31,32] study, where stool samples were obtained
from infants at days 30, 120, and 365 after birth. In this illustration, we
restricted to children who were vaginally delivered with no perinatal
antibiotics exposure and exclusively breastfed during the first four
months of life. We chose to study the correlations at 30, 120, and
365 days, with 46, 44, and 34 sequenced samples, respectively, as this
represents a period where the environment changes dramatically due
to changes in diet (breastfeeding, mixed, solid foods), external con-
tacts and activities and in which much of the programming of the

immune system takes place[38,39]. We analyzed data at the family level
(prevalences provided in Supplementary Table 5).

Bar graphs of temporal changes in the bias-corrected counts and
relative abundances of the top ten prominent families and phyla are
provided in Fig. 6 and Supplementary Fig. 10, respectively. Consistent
with the literature on the gut microbiome of breastfed infants, during
the first 30 days, we found a greater relative abundance of the phylum
Actinobacteria, specifically the family *Bifidobacteriaceae* (55%), along
with families of Firmicutes, such as *Clostridiaceae* (10%), *Lachnospir-*
*aceae* (1%), *Ruminococcaceae* (9%), and opportunistic *Staphylococca-*
*ceae* (1%). Since these babies were born vaginally, we also see some
*Bacteroidaceae* (6%) during the first 30 days. Pro-inflammatory bac-
terial families *Enterobacteriaceae* (9%) and *Pasteurellaceae* (2%), in the
phylum Proteobacteria are also present at this age. As the baby's
environment changes, in particular the transition from breastmilk to
other foods, we observe, as expected, that the relative abundance of
*Bifidobacteriaceae*, which is promoted by oligosaccharides in breast-
milk, temporally decreases towards the end of the first year (55, 35, 5%
at days 30, 120, and 365, respectively). We also notice changes in the
relative abundance of *Clostridiaceae* (10, 30, 3% at days 30, 120, and

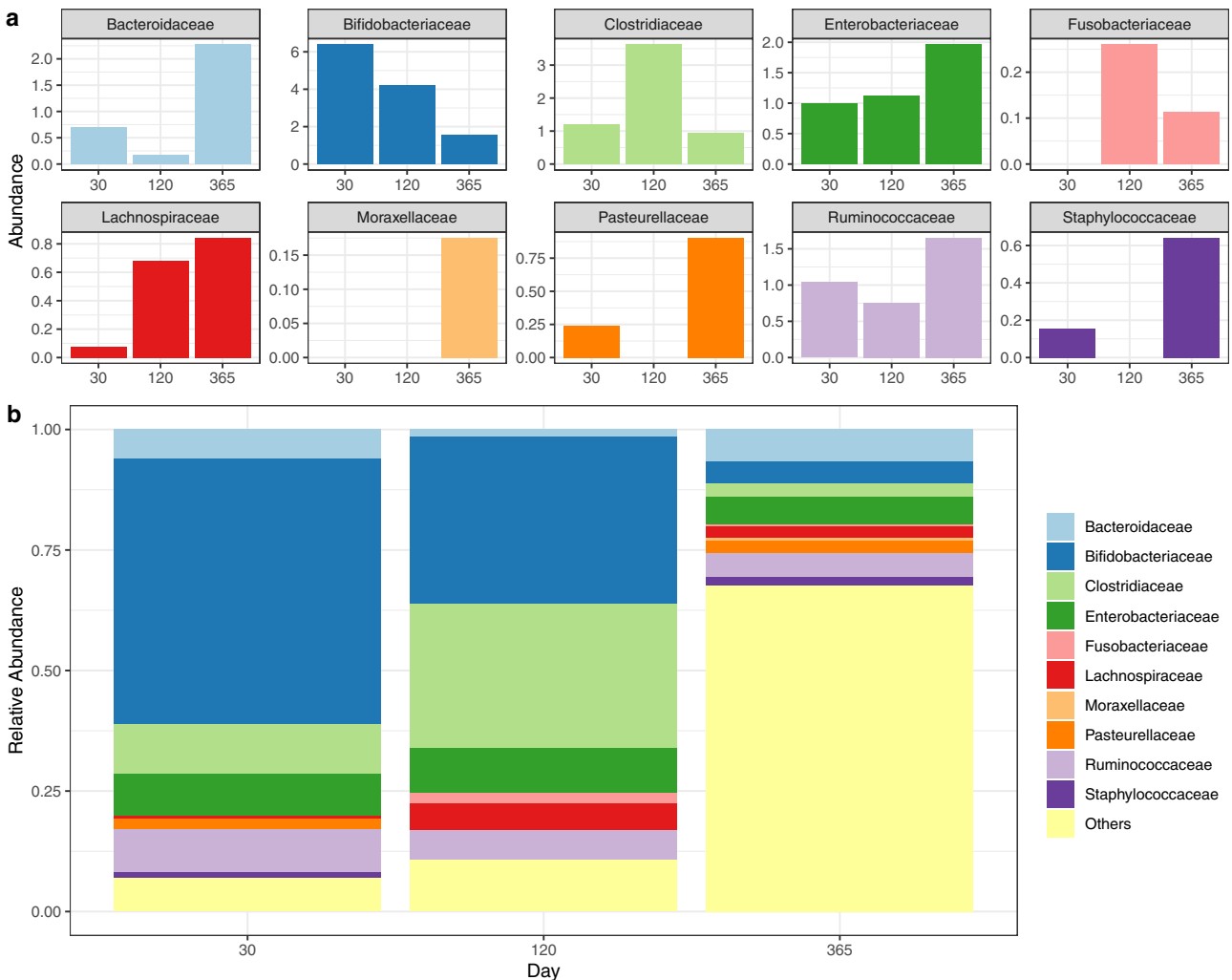

**Fig. 6 | Temporal patterns of bias-corrected abundance and relative abundances for Norwegian infant gut microbiome data among families.** The x-axis represents days (30, 120, and 365 days) and the y-axis denotes either the bias-corrected (**a**) abundances or **b** relative abundances. The top ten most abundant taxa up to 365 days were selected for the visualization purpose. Families with prevalence across samples <10% were not shown separately and grouped into "Others". Different taxa were coded by different colors, as shown on the legends.

365, respectively). The relative abundance of *Enterobacteriaceae* was constant throughout the first year (9, 9, 6% on days 30, 120, and 365, respectively). As babies begin to crawl, touch various objects, and are in contact with other people or the environment, we expect new families to appear. In line with this, we see the arrival of *Moraxellaceae* (1%) in the gut at Day 365 and numerous other families which are grouped as "Others" (68%).

To the best of our knowledge, this is the first paper to characterize correlations among different families of bacteria during the first year of development of a baby. The SECOM (Pearson2) and SECOM (Distance) correlations among the top ten prevalent families are summarized in Fig. 7. During the first 30 days of birth, we see significant negative linear correlations between *Enterobacteriaceae* and *Bifidobacteriaceae* (SECOM (Pearson2) = −0.44, SECOM (Distance) = 0.44) and between *Enterobacteriaceae* and *Bacteroidaceae* (SECOM (Pearson2) = −0.61, SECOM (Distance) = 0.61). We hypothesized that inter-relationships among the gut microbiota change with time as new microbes arrive. In line with this, we discovered strong negative linear relationships between the pathogenic family *Enterobacteriaceae* and several important commensal families such as *Lachnospiraceae* (SECOM (Pearson2) = −0.89, SECOM (Distance) = 0.92) and *Clostridiaceae* (SECOM (Pearson2) = −0.75, SECOM (Distance) = 0.73) at day 120. We also saw negative linear

correlation between *Bifidobacteriaceae* and *Clostridiaceae* (SECOM (Pearson2) = −0.49, SECOM (Distance) = 0) at day 120. By Day 365, as expected, many new families start colonizing the gut and developing complex interactions among them. On Day 365, as the (bias-corrected) abundance of phylum Actinobacteria decreases (Supplementary Fig. 10), in particular, due to the decrease in the abundance of *Bifidobacteriaceae*, *Lachnospiraceae* increases (Fig. 6a) and the two families are linearly negatively correlated with SECOM (Pearson2) = −0.45 (SECOM (Distance) = 0.5). *Lachnospiraceae* continues being negatively correlated with *Enterobacteriaceae* (SECOM (Pearson2) = −0.37, SECOM (Distance) = 0.42), and with opportunistic family *Staphylococcaceae* (SECOM (Pearson2) = −0.74, SECOM (Distance) = 0.78). *Lachnospiraceae* is also nonlinearly correlated with another important gut microbiota *Ruminococcaceae*, with SECOM (Distance) = 0.42. Similar to *Lachnospiraceae*, the *Ruminococcaceae* family is also negatively correlated with *Enterobacteriaceae* (SECOM (Pearson2) = −0.42, SECOM (Distance) = 0.45) and *Staphylococcaceae* (SECOM (Pearson2) = −0.56, SECOM (Distance) = 0.62). Not surprisingly, *Ruminococcaceae* family is positively linearly correlated with *Bacteroidaceae* with SECOM (Pearson2) = 0.34 (SECOM (Distance) = 0.37). We see a nonlinear relationship between *Ruminococcaceae* and *Bifidobacteriaceae* with SECOM (Distance) = 0.36. We also find a nonlinear relationship

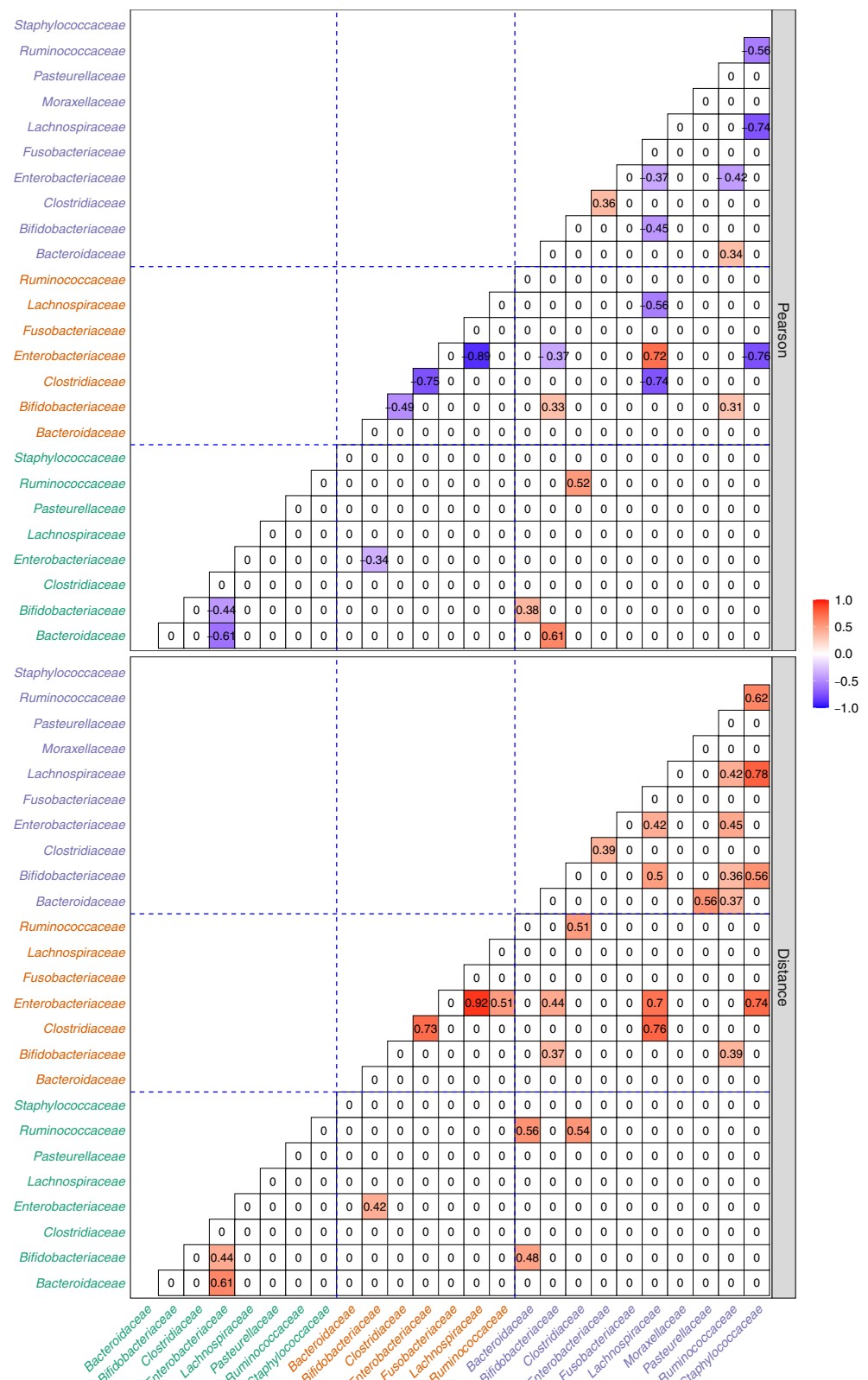

**Fig. 7 | Temporal patterns of correlations for Norwegian infant gut microbiome data among families.** The correlation measures (Pearson and Distance) are represented by rows. Families from days 30, 120, and 365 are colored in green, brown, and purple on the x and y axes, respectively. Blue dashed lines are used to separate families between different time points. The top ten most abundant families up to 365 days were selected for the visualization purpose. Families with prevalence across samples <10% were discarded. The correlation coefficient ranges from −1 to 1, color-coded by blue to red, respectively.

between *Bifidobacteriaceae* and the opportunistic family *Staphylococcaceae* with SECOM (Distance) = 0.56. A nonlinear relationship was seen between *Bacteroidaceae* and *Pasteurellaceae* with SECOM (Distance) = 0.56. A positive correlation between *Enterobacteriaceae* and *Clostridiaceae* of 0.36 was seen (SECOM (Distance) = 0.39). Thus, in most cases, we notice that the distance and Pearson correlation coefficients seem to track each other. However, there are instances where the relationships appear to be nonlinear with zero Pearson correlations. In addition to the intra-time correlations described above, we also computed pairwise inter-time or temporal correlations among days 30, 120, and 365. When it comes to pairwise temporal correlations, there are two sources of sparsity, namely, sparsity within time points and sparsity across time points, because samples were not available on every infant at all three time points. Secondly, as noted in Flores et al. [37], the temporal variability in measurements within a subject can be substantially large that it can overwhelm correlations between pairs of taxa over time. Thus, unlike correlations within a time point, we expect more diffused correlations across time points. Interestingly, despite these challenges, the SECOM methodology identified several pairs of taxa to be correlated across time points (Fig. 7). These correlations generate interesting hypotheses to investigate in the future.

In summary, we find that from 4 months of age, as the baby begins to ingest more complex foods, *Lachnospiraceae* and *Ruminococcaceae* interact with several families. These two families play an important role in degrading complex polysaccharides to short-chain fatty acids such as acetic, butyric, and propionic acid, which have numerous biochemical functions[40]. Thus, it is not surprising that during the growth and development of a baby during its first year, these families interact with many other families of bacteria to achieve a balance in the gut ecology. Not surprisingly, during the early days of gut colonization (Day 30 and Day 120), important families that promote infants' health, such as *Bifidobacteriaceae*, *Bacteroidiceae*, *Lachnospiraceae*, and *Clostridiaceae*, are all negatively correlated with *Enterobacteriaceae*, a family of bacteria that contains many familiar pathogens, such as *Salmonella*, *Shigella*, and *Escherichia coli*.

For comparison purposes, we also implemented SparCC on the above NoMIC data due to its popularity for computing correlations among microbiota. The results using SparCC are summarized in Supplementary Fig. 11. According to SparCC, none of the families are correlated with each other on days 30 and 120. Whereas SECOM discovered some biologically plausible correlations. On day 365, SparCC uniquely identified a nonzero correlation coefficient between *Moraxellaceae* and *Bifidobacteriaceae*; however, the correlation coefficient of −0.3 is at the margin of its threshold value of 0.3. On the other hand, SECOM identified correlations between 8 pairs of families that were not identified by SparCC. Note that, as in the "forehead" and "palm" example above, the correlation estimates based on SECOM (Pearson2) and SECOM (Distance) are very similar whenever SECOM (Pearson2) ≠ 0. In contrast to SECOM, very few taxa were discovered to be temporally correlated by SparCC, This finding is consistent with the findings in forehead and palm data as well as the simulations.

## Discussion

Estimation of correlations among microbes has several statistical challenges such as compositionality, excess of zeros, complex dependencies, high dimensionality, and sparsity. Under suitable assumptions, the SECOM methodology addresses these challenges. SECOM (Pearson1) and SECOM (Pearson2) are designed for estimating linear correlation coefficients, whereas SECOM (Distance) is designed for estimating nonlinear correlation coefficients.

In the case of linear relationships, we discovered that SECOM methods outperform SparCC by having higher accuracy (smaller relative loss), smaller FPR while being comparable in TPR. The

proportionality method had the smallest TPR in some cases and high FPR as well. Although SPIEC-EASI methods perform as well as SECOM in the above criteria, computationally, on average, they can be 100 times slower than SECOM. When it comes to nonlinear relationships, as expected, none of the methods competed well with SECOM (Distance) in terms of all criteria mentioned above.

Excess zeros in the data present a challenge for all correlation methods described in this paper. From our simulations, we see that for rare taxa with a large proportion of zeros (e.g., 90%), it may be better to perform a complete case analysis (CCA) rather than imputing zeros by adding pseudo-counts. Pseudo-counts can potentially either inflate or deflate correlation coefficients substantially (Supplementary Fig. 12). We see better results using CCA. If the proportion of zeros is small, then the SECOM model framework can be used for imputing the missing values. However, our simulation studies find the CCA method to perform just as well as when zeros were imputed (Supplementary Fig. 13).

SECOM accounts for compositionality by eliminating sample-specific and taxon-specific biases. This results in an easy-to-use and familiar linear model framework. This gives SECOM a distinct advantage over all other compositionally robust correlation methods, which are based on relative proportions of taxa when estimating correlations across multiple ecosystems. This is illustrated using simulated data as well as forehead and palm microbiome data obtained in Flores et al[37] and infant gut microbiome data obtained from the NoMIC study[31,32]. For instance, as one would expect, same genera from the two skin sites, forehead and palm, were found to be highly correlated by SECOM. However, to our surprise, none of the other methods found any of the genera to be correlated between the two skin sites.

Using the infant gut microbiome data obtained from the NoMIC study, for the first time in the literature, we developed a baseline understanding of naturally occurring correlations among various families of bacteria during the first year after birth. We limited our sample of babies to those who were born vaginally, breastfed for the first 4 months, and not exposed to antibiotics during the first year. During the first 30 days of birth, a baby's gut consists of very few families that are largely derived from maternal contact, including breastfeeding. By 4 months after birth, as the baby's contact with its environment increases, the bacterial diversity in the gut increases with new families of bacteria arriving. With babies' contact with their environment continuously changing, and as they wean from breastmilk to other forms of food, the gut microbial composition continuously evolves with the arrival of new families of gut bacteria. Thus, by the end of the first year, we expect more complex interactions among the families of gut bacteria. Using SECOM, we saw interesting temporal trends in correlations among various families, which seem to mirror temporal changes in the gut flora. For example, *Lachnospiraceae* and *Ruminococcaceae* are known to play an important role in the production of short-chain fatty acids which are also evolving during the first year of birth. We found these two families to correlate temporally with several families. Interestingly, as the babies wean from breastmilk to more solid foods by the end of the first year, we found *Lachnospiraceae* to be negatively correlated with *Bifidobacteriaceae*, which is largely obtained from breastmilk. We also discovered that at various time points, *Enterobacteriaceae*, a family that contains pathogens such as *Salmonella*, *Shigella*, and *Escherichia coli*, is negatively correlated with *Bacteroidiceae*, *Lachnospiraceae*, and *Clostridiaceae*. In the future, we would be interested in understanding how these relationships are modified when there are clinical interventions such as C-section and the use of antibiotics, which are common practices globally. Subsequently, we will also be interested in investigating the effects of these associations on various health outcomes among infants. A limitation of the present study is a small sample size, due to which we may have missed some important correlations.

## Methods

### Description of some existing methods

**Proportionality.** The $\phi$ statistic was proposed to describe the strength of proportionality between a pair of variables or taxa. It is closely related but not identical to the correlation coefficient. Instead, it is a "goodness-of-fit for proportionality" statistic to assess the extent to which a pair of taxa are proportional[15]. $\phi = 0$ indicates that a pair of taxa are exactly proportional (e.g., $y = ax$); on the other hand, $\phi \neq 0$ as long as a pair of taxa are not exactly proportional to each other, even if they are perfectly linearly correlated (e.g., $y = ax + b$). Additionally, unlike the standard Pearson correlation coefficient, the magnitude of $\phi$ is hard to interpret because it lacks a scale. For instance, by knowing $\phi = 0.1$ we cannot assess how strongly a pair of taxa are proportionally related to each other. Also, $\phi$ statistics is not symmetric ($\phi(x, y) \neq \phi(y, x)$), making it inappropriate to serve as a dissimilarity measure for downstream analyses such as ordination or clustering analysis. Lastly, there is no clear criterion to produce a sparse matrix based on $\phi$ statistics.

**Sparse correlations for composition data (SparCC)[14].** It is designed to estimate the correlation coefficients for log-transformed abundances while addressing bias due to compositionality in the data. It assumes that (1) the number of taxa is large and (2) the underlying true correlation matrix is sparse. However, similar to the proportionality method, it is not a statistically rigorous approach to obtain a sparse correlation matrix for SparCC. A cutoff of $r = 0.3$ was suggested in Friedman et al.[14] to declare if a pair of taxa are correlated or not. More accurate estimation can be achieved by iterating the basic SparCC inference procedure. At each iteration, the strongest correlated pair identified in the previous iteration is excluded, which reinforces sparsity among the remaining pairs and yields better correlation estimates.

**Sparse inverse covariance estimation for ecological association inference (SPIEC-EASI)[16].** It is a measure of association that exploits the concept of conditional independence and graphical models. Two taxa are conditionally independent if given the abundances of all other taxa, neither taxon provides additional information about the abundance of the other. Otherwise, these taxa are not conditionally independent and there is a linear relationship between them as the method is essentially estimating the inverse covariance matrix (precision matrix). While conceptually it is a useful measure, as one would expect, SPIEC-EASI is computationally very intensive compared to the above existing methods.

### SECOM model set-up

All notations used in the following are provided in Table 1. The absolute abundances $A_{ij}$ in a unit volume of an ecosystem are unobservable non-negative integers. Let $\mathcal{T}$ be the set of all possible configurations of the presence or absence of a taxon in a sample, then

$$\mathcal{T} = \begin{cases} (1, 1, \ldots, 1) \\ (0, 1, \ldots, 1), \ldots, (1, 1, \ldots, 0) \\ \quad\quad\quad \vdots \\ (0, 0, \ldots, 1), \ldots, (1, 0, \ldots, 0) \end{cases}$$

where 0 and 1 correspond to the cases where a taxon is missing (zero) or present in a sample, respectively. We represent each event in $\mathcal{T}$ by $T$ and it is indexed by $t = 1, \ldots, 2^d - 1$. For example, $T_1$ corresponds to the case where all taxa are present in a sample, and $T_{2^d-1}$ is the case where only the first taxon is present, and all remaining taxa are zeros.

We further assume that for each sample, the pattern of taxon presence is generated by independent random variables (r.v.) $M_{ij}$,

**Table 1 | Definitions of key notations**

| Term | Definition |
| --- | --- |
| $i$ | Sample, $i = 1, \ldots, n$. |
| $j$ | Taxon, $j = 1, \ldots, d$. |
| $A_{ij}$ | The absolute abundance in the ecosystem for taxon $j$ in sample $i$. |
| $O_{ij}$ | The observed abundance for taxon $j$ in sample $i$. |
| $S_i$ | Sample-specific sampling fraction. |
| $C_j$ | Taxon-specific sampling efficiency. |
| $E_{ij}$ | Random error for taxon $j$ in sample $i$. |
| $a_{ij}$ | $\log A_{ij}$. |
| $o_{ij}$ | $\log O_{ij}$. |
| $s_i$ | Sample-specific sampling fraction in log scale. |
| $c_j$ | Taxon-specific sequencing efficiency in log scale. |
| $e_{ij}$ | Random error for taxon $j$ in sample $i$ in log scale. |

$1 \leq i \leq n$, $1 \leq j \leq d$ such that

$$P(M_i = T_t) = \delta_t^i, 0 \leq \delta_t^i \leq 1,$$

where $M_i = (M_{i1}, \ldots, M_{id})^T$.

We propose that the observed abundance is generated by the following multiplicative model:

$$O_{ij} = S_i C_j A_{ij} E_{ij}. \tag{1}$$

For each $O_{ij}$ with $M_{ij} = 1$, a log transformation of the above multiplicative model results in the following additive model,

$$o_{ij} = s_i + c_j + a_{ij} + e_{ij}. \tag{2}$$

Conditional on $M_i = 1$, assume that the log of absolute abundance has finite first and second moments:

$$E(a_{ij}) = \alpha_j, Var(a_{ij}) = \sigma_{jj}^0. \tag{3}$$

Additionally, we assume that:

$$s_i \text{ is a sample specific parameter,}$$
$$c_j \text{ is a taxon specific parameter,}$$

$$e_{ij} \sim_{i.i.d.} F_e \text{ with } E(e_{ij}) = 0, Var(e_{ij}) = \sigma_e^2, \tag{4}$$

$$a_{ij} \perp\!\!\!\perp e_{ij}. \tag{5}$$

We define two concepts of correlation coefficients in this section. One quantifies the strength of the linear relationship between a pair of taxa by modifying the Pearson correlation coefficient for sparse high-dimensional compositional data. The second quantifies the strength of the nonlinear relationship between a pair of taxa by modifying the distance correlation coefficient.

**Remark 1.** Since the distance correlation coefficient is a measure of general dependency between a pair of taxa, it does not have a sign and is always non-negative. On the other hand, the Pearson correlation coefficient can take positive or negative values depending upon whether they are positively or negatively linearly correlated.

**Remark 2.** For clarity, in this remark, we spell out the hierarchy among the various types of dependencies between a pair of taxa and various concepts of correlation coefficients.
1.  Hierarchy of dependencies

- {Linear relationship} ⊂ {Monotonic relationship} ⊂ {Non-linear relationship} ⊂ Statistical dependence
2. Hierarchy of measures of dependence
   - Pearson correlation coefficient: quantifies linear dependence
   - Spearman correlation coefficient: quantifies monotonic dependence
   - Distance correlation coefficient: quantifies statistical dependence

The simplest of dependencies between two taxa is linear, and the Pearson correlation coefficient is designed to quantify this relationship. The Spearman correlation coefficient is designed to quantify monotonic relationships. Since the linear relationship is a special case of monotonic relationships, the Spearman correlation coefficient may also be used to quantify the linear relationship. Statistical dependence between two taxa includes nonlinear, non-monotonic relationships, monotonic, as well as simple linear relationships. Therefore the distance correlation coefficient, which is designed to quantify statistical dependence between two taxa, can be used to quantify any dependence, whether linear, monotonic, non-monotonic, or nonlinear relationships. Furthermore, in theory, if the distance correlation coefficient is zero, then we conclude statistical independence between two taxa. This implies a lack of any of the above relationships. If the Spearman correlation coefficient is zero, then it implies a lack of monotonic relationship, which also implies a lack of linear relationship. If the Pearson correlation coefficient is zero, then one can only infer a lack of linear dependence, but other forms of relationships cannot be ruled out.

**Remark 3.** In the simulation studies and real data applications presented in the main text, SECOM methodology outputs results either based on the Pearson correlation measure (namely, SECOM (Pearson1) and SECOM (Pearson2)) or distance correlation measure (SECOM (Distance)). If the relationship between a pair is nonlinear, then we expect SECOM (Distance) ≠ 0, whereas SECOM Pearson methods could be zero. On the other hand, if the relationship is linear, then the SECOM Pearson methods are expected to be non-zero, and the sign of the coefficient is positive or negative depending upon whether the two taxa are positively or negatively associated. When SECOM Pearson methods are nonzero, then we expect SECOM (Distance) ≠ 0 as well. This is because a linear relationship is a special case of dependency. In such cases, we recommend the user use the value provided by the SECOM Pearson correlation coefficients. However, there may be some rare instances where SECOM Pearson correlation coefficients are nonzero, but the distance correlation is zero. This can happen due to a couple of reasons, (a) small sample size and the sparsity constraints or (b) the nonzero SECOM Pearson correlation coefficients are spurious. In such cases, we recommend the researcher use the SECOM Pearson correlation coefficient with caution.

**Remark 4.** If one's research interest is identifying linear relationships, and the sample size is generally large (>50 based on our simulation benchmarks), then the thresholding method (SECOM (Pearson1)) is recommended for the sparsity of the estimated correlation matrix as it is theoretically proved to be a consistent estimator. On the contrary, if identifying general dependencies (linear and nonlinear relationships) between taxa is the primary purpose, or the sample size is limited, then the $p$ value filtering method (SECOM (Pearson2) or SECOM (Distance)) is recommended for sparsity.

**Remark 5.** A common problem with microbiome data is the presence of excess zeros which present challenges when computing

correlation coefficients. This is an issue for all methods in the literature. SECOM implements the strategy of a complete case analysis (CCA) to handle zeros, i.e., uses only pairs of nonzero data. However, as an alternative strategy to CCA, one may want to add a pseudo-count, e.g., 1 (Add One), before computing correlation coefficients. We conducted a simulation study (Supplementary Fig. 12) to evaluate these two strategies. We simulated a complete pair of true abundances using a negative-binomial distribution. Using these complete data, we created sparsity by either forcing some of the true abundances to be zeros (structural zeros[41], Supplementary Fig. 12a, b) or by multiplying small sampling fractions to the true abundances and rounding decimals to zeros (sampling zeros[41], Supplementary Fig. 12c, d). We applied CCA and Add One strategies to the resulting data and found that when there were structural zeros between a pair of taxa, then adding pseudo-counts could either inflate or deflate a true correlation coefficient. On the other hand, the correlation coefficient based on the CCA was closer to the estimate based on the full non-missing data (Supplementary Fig. 12a, b). If zeros were caused by small sampling fractions, either using complete cases or adding a pseudo-count yielded similar results (Supplementary Fig. 12c, d).

SECOM, analogous to ANCOM-BC[7], is a de-biasing model. Hence, after correcting for the sample-specific and taxon-specific biases, the model reduces to a traditional linear model. Consequently, it enables the user to impute missing values to calculate correlation coefficients. We conducted an exhaustive simulation study where the percent of missing values ranged from 0 to 70% for two taxa generated from log-normal distributions. We compared CCA with two different imputing strategies, namely, Gradient Boosting Machine (GBM)[42,43] and Multivariate Imputation by Chained Equations (MICE)[44] by calculating standard Pearson correlation coefficients on complete data and imputed data, respectively. We found that there was not much difference between the three estimates (Supplementary Fig. 13), and CCA-based estimates compete well with those obtained from GBM or MICE imputed data.

**Remark 6.** Researchers may be interested in computing correlation coefficients within and between two or more ecosystems using samples obtained on the same subject. For example, one may be interested in correlating microbial abundances in two body sites, say "forehead" and "palm". In such cases, one concatenates data matrices from two ecosystems to estimate correlations between body sites and within body sites. Methodologies that honor compositionality structure in the data, would convert all observed abundances into relative abundances in the concatenated data and then compute correlation coefficients. While this approach appears to be reasonable, the correlation estimates within a body site, say palm, obtained from the concatenated data may be substantially different from the estimates one would have gotten if they estimated correlations within each site separately. This lack of coherence between the two estimation procedures for the same statistical parameter is unsatisfactory. It is a direct consequence of deriving relative abundances from the combined data and then calculating correlations. SECOM does not suffer from this problem because it first corrects the sample and taxon-specific biases and then applies the methodology described in this paper. In doing so, it provides coherent site-specific correlations.

**Remark 7.** As with any statistical procedure, SECOM methodology makes some assumptions regarding the data and the problem as follows:
- There are two major sources of bias in the observed count data, namely, sample-specific sampling fraction and taxon-specific sequencing efficiency. These sources of bias are now well

accepted in the microbiome literature[18]. SECOM methodology corrects these biases.

- The correlation matrix is a sparse matrix. This is a very common assumption that is necessary when estimating correlation and covariance matrices in high dimensions. Basically, any method for estimating a $p \times p$ correlation matrix using a small sample size $n$, where $n < p$, results in an estimator that is mathematically a singular matrix, although the underlying true correlation matrix is a positive definite matrix. Consequently, the correlation matrix cannot be unbiasedly or consistently estimated by any method unless some sparsity conditions are imposed on the correlation matrix, i.e., some of the true correlation coefficients need to be constrained by zero.

- Large signal-to-noise ratio. This is a very common assumption in classical linear regression analysis. If the variance of the regression model is substantially large compared to the regression coefficient, then statistical inferences on the regression parameter will be underpowered unless the sample sizes are very large. Thus, in all regression analyses, to have high power, one implicitly requires large effect sizes (signal-to-noise ratio). A similar phenomenon holds in the present situation. Although, for some types of data, this ratio may be empirically estimated from the data, it has to be taken as a regularity condition like any other condition in classical asymptotic theory.

- Empirical joint probability of observing a pair of taxa is nonzero. In other words, we avoid situations where a pair of taxa are missing in all samples because, in that case, it is not possible to compute the correlation coefficient between them.

## Estimation of linear correlation coefficients

In this section, we develop a methodology for estimating the linear correlation coefficient between the abundances of pairs of taxa $l$ and $m$, $1 \le l, m \le d$, while noting that for each subject $i$, the abundances $a_{il}$ and $a_{im}$ are not observable. The only observable values are the observed counts, e.g., OTUs or ASVs, $o_{il}$ and $o_{im}$. According to our hypothesized model, the actual abundance $a_{ij}$ of the $j$th taxon in a unit volume of the specimen from the $i$th subject, in expectation, is a function of observed counts $o_{ij}$, and unknown nuisance parameters $s_i$ and $c_j$. Therefore, to obtain an approximate unbiased estimator of $a_{ij}$ using the observable $o_{ij}$, we need to estimate nuisance parameters $s_i$ and $c_j$ and eliminate the bias introduced by them. Once that is done, we can define Pearson or Spearman correlations using the modified observed counts $o_{ij}$ to derive bias-corrected estimates of the true correlation coefficient between the abundance of a pair of taxa. In the following, we now describe the methodology to eliminate the nuisance parameters $s_i$ and $c_j$.

Denote the covariance matrix of $a_{ij}$ as $\Sigma^0 = [\sigma_{lm}^0]_{l,m=1,\dots,d}$, and the Pearson correlation coefficient matrix as $R^0 = [\rho_{lm}^0]_{l,m=1,\dots,d}$. We make the following sparsity assumption regarding the covariance matrix.

**Assumption 1.** Approximately sparse covariance matrices

$$\mathcal{U}(K) = \left\{ \Sigma : \sigma_{jj} < K, \frac{1}{d^2}\sum_{l \ne m}\sigma_{lm} = o(1) \right\}.$$

Two examples of matrices that satisfy this condition are (1) a d-diagonal matrix and (2) the AR(1) covariance matrix.

From (2), (3), (4), and (5), by centering the data across samples, we eliminate the effect of taxon-specific sampling efficiency $c_j$ as follows:

$$o_{ij} - \bar{o}_{.j} = (s_i - \bar{s}_.) + (a_{ij} - \bar{a}_{.j}) + (e_{ij} - \bar{e}_{.j}) = ① + ② + ③.$$

where, for the $j$th taxon and $i = 1, 2, .., n$, $\bar{s}_. = \frac{1}{n}\sum_{i=1}^n s_i$, $\bar{a}_{.j} = \frac{1}{n}\sum_{i=1}^n a_{ij}$, $\bar{e}_{.j} = \frac{1}{n}\sum_{i=1}^n e_{ij}$. Note that:

(1) We call ① the "sampling fraction difference", which is a sample-specific bias term,
(2) For ②: $E(a_{ij} - \bar{a}_{.j}) = 0, \mathrm{Var}(a_{ij} - \bar{a}_{.j}) = (1 - \frac{1}{n})\sigma_{jj}^0$,
(3) For ③: $E(e_{ij} - \bar{e}_{.j}) = 0, \mathrm{Var}(e_{ij} - \bar{e}_{.j}) = (1 - \frac{1}{n})\sigma_e^2$.

Define the set of nonzero taxa in sample $i$ by $d(i) = \{j : M_{ij} = 1, 1 \le j \le d\}$. Then by on Assumption 1, we have

$$\frac{1}{|d(i)|}\sum_{j \in d(i)}\left\{(o_{ij} - \bar{o}_{.j}) - ①\right\} = \frac{1}{|d(i)|}\sum_{j \in d(i)}② + \frac{1}{|d(i)|}\sum_{j \in d(i)}③ \longrightarrow_p 0 \text{ as } |d(i)| \to \infty. \quad (6)$$

Therefore, a quick and simple estimator of the sampling fraction difference is the arithmetic mean of the difference between observed abundances, i.e., $\widehat{①} = \widehat{s_i - \bar{s}_.} = \frac{1}{|d(i)|}\sum_{j \in d(i)}(o_{ij} - \bar{o}_{.j})$. Taking into account the variance of residuals $o_{ij} - \bar{o}_{.j}$, a weighted average is proposed as follows:

$$\widehat{①} = \widehat{s_i - \bar{s}_.} = \sum_{j \in d(i)} w_j(o_{ij} - \bar{o}_{.j}), \quad (7)$$

where $w_j = \frac{1}{\mathrm{Var}(o_{ij} - \bar{o}_{.j})}\Big/\sum_{l \in d(i)}\frac{1}{\mathrm{Var}(o_{il} - \bar{o}_{.l})}$.

Denote the centered absolute abundance by $x_{ij} = a_{ij} - \bar{a}_{.j}$, $1 \le i \le n$, $1 \le j \le d$. Then

$$\mathrm{Cov}(x_{il}, x_{im}) = \left(1 - \frac{1}{n}\right)\sigma_{lm}^0, \mathrm{Var}(x_{ij}) = \left(1 - \frac{1}{n}\right)\sigma_{jj}^0, \mathrm{Corr}(x_{il}, x_{im}) = \rho_{lm}^0.$$

Similarly, define the centered absolute abundance with noise by

$$y_{ij} := (a_{ij} - \bar{a}_{.j}) + (e_{ij} - \bar{e}_{.j}) = x_{ij} + (e_{ij} - \bar{e}_{.j}) = (o_{ij} - \bar{o}_{.j}) - (s_i - \bar{s}_.),$$

and denote $\mathrm{Cov}(y_{il}, y_{im}) := \sigma_{lm}, \mathrm{Corr}(y_{il}, y_{im}) := \rho_{lm}, \Sigma = [\sigma_{lm}]_{l,m=1,\dots,d}$, $R = [\rho_{lm}]_{l,m=1,\dots,d}$. Then the off diagonal and diagonal elements of $\Sigma$ are, respectively,

$$\sigma_{lm} = \left(1 - \frac{1}{n}\right)\sigma_{lm}^0, \sigma_{ll} = \left(1 - \frac{1}{n}\right)(\sigma_{ll}^0 + \sigma_e^2).$$

We make the following assumption that noise-to-signal ratio is very small.

**Assumption 2.** Small noise-to-signal ratio

$$\sigma_e^2 \ll \sigma_{ll}^0.$$

Then we have:

$$\rho_{lm} = \frac{\sigma_{lm}}{\sqrt{\sigma_{ll}\sigma_{mm}}} = \frac{\sigma_{lm}^0}{\sqrt{(\sigma_{ll}^0 + \sigma_e^2)(\sigma_{mm}^0 + \sigma_e^2)}} \approx \frac{\sigma_{lm}^0}{\sqrt{\sigma_{ll}^0\sigma_{mm}^0}} = \rho_{lm}^0,$$

which means the Pearson correlation coefficient among $y's$ provides a good approximation to the Pearson correlation coefficient among $x's$ as well as among $a's$. Since $y_{ij} = (o_{ij} - \bar{o}_{.j}) - (s_i - \bar{s}_.)$. From (7), we estimate $y_{ij}$ by $\hat{y}_{ij} = (o_{ij} - \bar{o}_{.j}) - \widehat{s_i - \bar{s}_.}$. Thus, from (6),

$$\hat{y}_{ij} \longrightarrow_p (o_{ij} - \bar{o}_{.j}) - (s_i - \bar{s}_.) = y_{ij}, \text{ as } |d(i)| \to \infty. \quad (8)$$

In the sequel, for all $1 \le l, m \le d$, denote $n(l) = \{i : M_{il} = 1, 1 \le i \le n\}$ and $n(l, m) = \{i : M_{il} = M_{im} = 1, 1 \le i \le n\}$, where $n(l)$ is the number of samples where $l$th taxon is observed, and $n(l, m)$ represents the number of samples where both $l$th and $m$th are observed. For each $1 \le l$,

$m \le d$, let

$$\hat{\mu}_l = \frac{1}{|n(l)|} \sum_{i \in n(l)} \hat{y}_{il},$$

$$\hat{\sigma}_{lm} = \frac{1}{|n(l,m)|} \sum_{i \in n(l,m)} (\hat{y}_{il} - \hat{\mu}_l)(\hat{y}_{im} - \hat{\mu}_m),$$

$$\sum_n = [\hat{\sigma}_{lm}]_{l,m=1\ldots,d},$$

$$R_n = [\hat{\rho}_{lm}]_{l,m=1\ldots,d} = \left[\frac{\hat{\sigma}_{lm}}{\sqrt{\hat{\sigma}_{ll}\hat{\sigma}_{mm}}}\right]_{l,m=1\ldots,d}.$$

To prove the consistency of the proposed correlation estimator, we make the following mild Assumption 3 that each taxon (Assumption 3(i)) or each pair of taxa (Assumption 3(ii)) is present with a nonzero probability ($\delta_{\min}$).

**Assumption 3.** Minimal probability of presence. Define:

$$C_i(l) = \{t : \text{taxon } l \text{ is present in } T_t \text{ in } M_i, 1 \le t \le 2^d - 1\},$$

$$C_i(l,m) = \{t : \text{taxa } l \text{ and } m \text{ are present in } T_t \text{ in } M_i, 1 \le t \le 2^d - 1\}.$$

There exists a constant $\delta_{\min} > 0$ such that for any $1 \le l, m \le d$,

$$(i)\frac{1}{n}\sum_{i=1}^{n}\sum_{t \in C_i(l)} \delta_t^i = \delta_l > \delta_{\min}, (ii)\frac{1}{n}\sum_{i=1}^{n}\sum_{t \in C_i(l,m)} \delta_t^i = \delta_{l,m} > \delta_{\min}.$$

**Theorem 1.** Suppose $\Sigma^0$ satisfies Assumption 1 and Assumption 3 is valid, then

$$\|R_n - R\|_\infty = O_p\left(\sqrt{\frac{\log d}{n}}\right).$$

Thus, the empirical correlation matrix $R_n$ based on $\hat{y}'s$ consistently estimates the correlation matrix $R$ of $y_{ij}$ in the *Sup* norm. Under the Assumption 2, $R$ is a reasonable approximation to the true correlation matrix $R^0$. The details of the proof are provided in the Supplementary Methods.

Instead of Pearson correlation coefficient, if the Spearman correlation coefficient is of interest, then we convert $x_{ij}, y_{ij}, \hat{y}_{ij}$ into ranks $r_{ij}^0, r_{ij}, \hat{r}_{ij}$. Clearly, for $r_{ij}^0, r_{ij}, \hat{r}_{ij} \in \{1, 2, \ldots, n\}$,

$$\hat{r}_{ij} \longrightarrow_p r_{ij}, \text{ as } |d(i)| \to \infty, \quad (9)$$

$$r_{ij} \approx r_{ij}^0 \text{ as long as Assumption 2 holds.} \quad (10)$$

With a slight abuse of notations, denote $\text{Corr}(r_{il}, r_{im}) = s_{lm}$, $\text{Corr}(\hat{r}_{il}, \hat{r}_{im}) = \hat{s}_{lm}$, $S = [s_{lm}]_{l,m=1,\ldots,d}$, $S_n = [\hat{s}_{lm}]_{l,m=1,\ldots,d}$. Then we have the following result.

**Theorem 2.** Suppose $\Sigma^0$ satisfies Assumption 1 and Assumption 3 is true, then:

$$\|S_n - S\|_\infty = O_p\left(\sqrt{\frac{\log d}{n}}\right).$$

The proof of this result relies on the observation that standardized rank $\zeta_{il} = \frac{\hat{r}_{il} - E(\hat{r}_{il})}{\sqrt{\text{Var}(\hat{r}_{il})}}$ is naturally bounded, which allows the applicability of the Hoeffding's inequality. The details of the proof are deferred to the Supplementary Methods.

When computing correlations of high-throughput sequencing data, it is reasonable to assume that only a small fraction of features (genes/proteins/taxa, etc.) are correlated with each other, i.e., the correlation matrix is sparse. Inspired by the Sparse Estimation of the Correlation matrix (SEC) approach[45], we estimate the sparse correlation matrix by solving

$$\hat{R} = \arg\min_R \frac{1}{2} \|R - R_n\|_F^2 + \lambda |R|_1, \text{ s.t.} R \succeq \epsilon I, R_{jj} = 1, j = 1, \ldots, d. \quad (11)$$

where
  (1)  $\|\cdot\|$ is the Frobenius norm,
  (2)  $R \succeq \epsilon I$ means that $R - \epsilon I$ is semipositive definite,
  (3)  $\lambda$ is the regularization parameter for the $l_1$ norm.

Denote the non-diagonal support of $R$ by $A_0 = \{l, m : l \ne m, \rho_{lm} \ne 0\}$ and its cardinality by $q$. The convergence of $\hat{R}$ to $R$ in the Frobenius norm is established in the following theorem.

**Theorem 3.** Suppose $\Sigma^0$ satisfies Assumption 1 and Assumption 3 is valid, and $\lambda = M\sqrt{\frac{\log d}{n}}$ for some constant $M$, we have the estimator defined in (11) such that

$$\|\hat{R} - R\|_F^2 = O_p\left(q\frac{\log d}{n}\right)$$

A sketch of the proof is provided in the Supplementary Methods.

From the computational point of view, the penalized likelihood problem stated in (11) can be replaced with the soft thresholding operator

$$\hat{R} = \text{sign}(R_n)(|R_n| - \lambda)_+, \quad (12)$$

where the tuning parameter $\lambda$ is selected by cross-validation $\lambda = \min_\lambda \|\hat{R} - R_n\|_F$.[23]

Since the solution $\hat{R}$ may not be symmetric in general, the final estimate of $\hat{R}$ is obtained by forcing $\hat{\rho}_{ll} = 1$ and choosing $\hat{\rho}_{lm} = \hat{\rho}_{lm}I(\hat{\rho}_{lm} \le \hat{\rho}_{ml}) + \hat{\rho}_{ml}I(\hat{\rho}_{lm} > \hat{\rho}_{ml})$. $\hat{R}$

The sparse matrix $\hat{R}$ can also be obtained by thresholding on the basis of $p$ values. Let $p_{lm}$ be the corresponding $p$ value for $\hat{\rho}_{lm}$, then $\hat{R}$ can be defined as

$$\hat{R} = [\hat{\rho}_{lm}I(p_{lm} < \alpha)]_{l,m=1,\ldots,d}, \quad (13)$$

where $\alpha$ is a pre-specified threshold, e.g., $\alpha = 0.005$. $\hat{R}$

### Estimation of a nonlinear correlation coefficient
In addition to linear correlation, we are also interested in describing nonlinear correlations among taxa within an ecosystem (e.g., gut or oral cavity) or across ecosystems. Therefore, the concept of distance correlation[34] was adopted to quantify general dependence between a pair of taxa.

**Definition 1.** Distance correlation coefficient (ref. 34). Let $X$ and $Y$ denote two random variables. Then the distance covariance and correlation coefficient between $X$ and $Y$ are defined as follows
  (1)  $d\text{Cov}^2(X,Y) = \frac{1}{\pi^2} \iint \frac{\|f_{X,Y}(t,s) - f_X(t)f_Y(s)\|^2}{t^2 s^2} dt ds,$
  (2)  $d\text{Var}^2(X) = \frac{1}{\pi^2} \iint \frac{\|f_{X,X}(t,s) - f_X(t)f_X(s)\|^2}{t^2 s^2} dt ds,$
  (3)  $d\text{Cor}(X,Y) = \frac{d\text{Cov}(X,Y)}{\sqrt{d\text{Var}(X)d\text{Var}(Y)}}.$

where $\|\cdot\|$ denotes Euclidean norm, and
  (1)  $0 \le d\text{Cor}(X, Y) \le 1$,
  (2)  $d\text{Cor}(X, Y) = 0$ if and only if $X$ and $Y$ are independent,
  (3)  $d\text{Cor}(X, Y) = 1$ implies that dimensions of the linear subspaces spanned by $X$ and $Y$ samples, respectively are almost surely equal.

Estimator of the distance correlation coefficient based on random samples $X = (X_1, X_2, \ldots, X_n)'$ and $Y = (Y_1, Y_2, \ldots Y_n)'$ is derived as follows. Define pairwise distances as $d_{ij} = |x_i - x_j|$, $e_{ij} = |y_i - y_j|$, and doubly centered distances as $D_{ij} = d_{ij} - \bar{d}_{i\cdot} - \bar{d}_{\cdot j} + \bar{d}_{\cdot\cdot}$, $E_{ij} = e_{ij} - \bar{e}_{i\cdot} - \bar{e}_{\cdot j} + \bar{e}_{\cdot\cdot}$, $i \neq j$, the empirical estimates of distance covariance and correlation can be obtained by

(1) $dCov_n^2(X, Y) = \frac{1}{n^2} \sum_{i=1}^n \sum_{j=1}^n D_{ij} E_{ij}$,

(2) $dVar_n^2(X, Y) = \frac{1}{n^2} \sum_{i=1}^n \sum_{j=1}^n D_{ij}^2$.

Note that the rank-transformed data also preserve the general dependence on the original data. We illustrate this using both linear and quadratic examples, as shown in Supplementary Fig. 14a, b, respectively. In Supplementary Fig. 14a, $y = 2x + e$, $x \sim U[-2, 2]$, $e \sim N(0, 1)$; whereas in Supplementary Fig. 14b, $y = x^2 + e$, $x \sim U[-2, 2]$, $e \sim N(0, 1)$. It is easy to see from the figure that even though the shape of $(X, Y)$ is not perfectly recovered using ranks (denoted as $(r(X), r(Y))$), distance correlations computed using the original data and the rank-transformed data (denoted as $dCor(X, Y)$ and $dCor(r(X), r(Y))$, respectively) are close to each other. This indicates that the distance correlation computed from the rank-transformed data provides a reasonably good approximation for the distance correlation computed from the original data. Therefore, researchers can use either original abundances or rank-transformed abundances to calculate distance correlations.

Similar to the procedure shown in the estimation of linear correlation coefficients, define

$$d_{lm} := dCor(y_{il}, y_{im}) \ (\text{or } dCor(r_{il}, r_{im})),$$
$$\hat{d}_{lm} := dCor(\hat{y}_{il}, \hat{y}_{im}) \ (\text{or } dCor(\hat{r}_{il}, \hat{r}_{im})),$$
$$D = [d_{lm}]_{l, m = 1, \ldots, d},$$
$$D_n = [\hat{d}_{lm}]_{l, m = 1, \ldots, d},$$

the sparse distance correlation matrix estimation can be obtained by thresholding or $p$ values:

$$\hat{D} = \text{sign}(D_n)(|D_n| - \lambda)_+, \text{or} \tag{14}$$

$$\hat{D} = [\hat{d}_{lm} I(p_{lm} < \alpha)]_{l, m = 1, \ldots, d}. \tag{15}$$

## Reporting summary
Further information on research design is available in the Nature Research Reporting Summary linked to this article.

## Data availability
Simulation data can be found on the corresponding GitHub repository[46]. The forehead and palm data can be found in Qiita: forehead https://qiita.ucsd.edu/study/description/2150 and palm https://qiita.ucsd.edu/study/description/2149. The NoMIC data used in this study are not publicly available but may be obtained by contacting Dr. Merete Eggesbø (Merete.Eggesbo@fhi.no).

## Code availability
SECOM has been implemented in the R package ANCOM-BC, which is available on Bioconductor at https://www.bioconductor.org/packages/release/bioc/html/ANCOMBC.html. All analyses shown in the paper can be found on the corresponding GitHub repository[46].

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

## Acknowledgements
We are grateful to all HUMIS cohort participants. The research of S.D.P. and H.L. was funded by the Intramural Research Program of the *Eunice Kennedy Shriver* National Institute of Child Health and Human Development (NICHD), National Institutes of Health, Bethesda, MD, USA. The research of M.E was funded in part by the Research Council of Norway, NEVRINOR [grant agreement no. 226402], Norway.

## Author contributions
Theory and Methodology were conceived and developed by H.L. and S.D.P. Illustration section was developed by H.L., M.E., and S.D.P. Interpretation of the results from the analysis of NoMIC study was performed by M.E. and S.D.P. All numerical works were carried out by H.L. and M.E. All three authors contributed to writing the paper.

## Funding

## Competing interests
The authors declare no competing interests.
