## [Peer Review File · Nature Communications]

REVIEWER COMMENTS

Reviewer #1 (Remarks to the Author):

This paper SECOM proposed a sparse estimation approach to infer correlations, both linear and nonlinear, between pairs of taxa in complex microbiome community. The authors allowed for two options, with one option using Pearson/Spearman correlation which capture the linear correlation and one option using distance correlation which captures the nonlinear correlation.

1. The model allows for the sampling fraction and sampling efficiency to differ between samples and taxa (S_i and C_j in the model), which is very appealing. These can be considered bias factors which act multiplicatively on the read counts. However, it seems that these factor only apply to the non-zero element in the count table. In other words, if a taxon has zero read count, the model assumes the zero count is accurate and S_i and C_j do not impact them. This is a limitation. You can easily imagine a situation that a taxon is in fact present in the sample, but if sequenced with low sampling efficiency, we observe zero read count for this taxon, yet if sequenced with high sampling efficiency, we observe non-zero read count. This can be problematic especially when more than 90% of the reads can be zeros. Could the authors elaborate this point, and/or assess the sensitivity of the proposed model against violations of such a simplification?
2. The authors provided two approaches for selecting important linear correlations, either through the “Thresholding” or “filtering” approach. However, it is not clear how the hypothesis testing (for calculating the p-values) is conducted, and how the “thresholding” is conducted.
3. How sensitive is the method to the choice of pseudo-counts, as the log transformation was applied to read counts?
4. Is there a theoretical guarantee of FDR of the proposed method on the identified correlations? What would be the assumptions that
5. The proposed method models the read counts instead of proportions with S_i and C_j to capture the efficiency of sequencing at the sample and taxon level. Is the model consistent with a compositional model, i.e., a model that works on relative abundances, and only has one set of vectors C_j to capture the efficiency in sequencing each taxon?
6. The extension of the model in handling multiple ecosystems is not very clearly described.

Some minor issues

1. supplemental figure 2 doesn't seem to make sense. Are both X and Y-axis log-transformed relative abundances? The number of samples are smaller than what was described in the paper. Further, there seems to have over-fitting for the LOESS curve: there was no point between 0.5 and 1, and how could the loess curve have a peak around $x = 1.5$?
2. There are mathematical assumptions behind the method and the theorems underlying it. It would be helpful if the readers can get some intuitive interpretation of these assumptions?

Reviewer #2 (Remarks to the Author):

This paper presents the method SECOM, which is a compositionally coherent approach to estimating linear (Pearson) correlation and nonlinear (distance) correlation in microbiome studies. The method is well-developed and will be useful to practitioners interested in more generic forms of association. The data application was particularly interesting - it's a very nice example of the utility of the method, as well as presenting some interesting biological conclusions. Overall, SECOM seems like a statistically valid and practically useful method.

Major Comments

1. I was confused for quite a while about how the settings in the paper translate to the context of a human microbiome study -- what is an "ecosystem" in a human microbiome study: a sample? a group of samples, perhaps defined by some phenotype or disease? Are analyses within an ecosystem just analyses of samples who all belong to the same group?
 - It would be helpful to provide some more explicit examples early in the paper and translate them into the language you use for the rest of the paper to clarify this for readers.
2. When you discuss comparison methods, one of the critiques is that introduction of sparsity is done in a relatively arbitrary/ad hoc way. But isn't a cutoff of, say, 0.3 in SPARCC conceptually similar to your filtering method, where you choose a cutoff of alpha and set values below that to zero (although SPARCC's version is based on correlation value rather than P-value)? You also apply the Filtering approach directly to the proportionality statistic (and P-value filtering is not entirely a new idea). I'm not sure whether this is as strong a differentiator between your method and previous methods as is implied (although you do see some improvement over SPARCC's method).

- Relatedly, is your statement that SPARCC recommends a cutoff of 0.3 based on the sentence from their manuscript “For clarity, only edges corresponding to correlations whose magnitude is greater than 0.3 are drawn.”? They also adapt the correlation estimation method to “reinforce sparsity” (their phrasing) – for accuracy, might be worth briefly mentioning that it’s not a standard correlation estimate when you describe their method on page 7.

3. In the Discussion, it is pretty well established by now that Pearson and Spearman correlation coefficients don’t work with microbiome data. In my opinion, the more interesting comparators for your method are SparCC and SPIEC-EASI, which were designed to handle microbiome data challenges and have ways to introduce sparsity. I’d prefer to see the takeaways related to these methods highlighted in the Discussion. If I understand your results correctly, the argument you’d make in that case is:

- SECOM is better at recovering sparsity patterns than SparCC
- Although SECOM and SPIEC-EASI perform similarly for TPR, FPR etc. in the case of linear associations, SPIEC-EASI is more computationally intensive – and it’s not great at sparsity recovery with nonlinear associations.

(And if I haven’t understood correctly, I’m doubly interested in seeing this paragraph in the Discussion!)

4. Simulations:

- True correlation among 50 out of 100 taxa does not seem particularly sparse to me. (Maybe your equation on page 2 is not accurately representing your sparsity pattern? That equation suggests taxon 1 is correlated with 2, 3, 4, ..., 50, and similarly for all other taxa in the first 50. The text would make me think taxon 1 was correlated with taxon 2, 2 with 1 and 3, 3 with 2 and 4, and so on.) SparCC says its sparsity assumption is strongly violated if >30% of pairs are correlated. If it’s really the case that the first 50 taxa are all associated, it might be worth doing some simulations with higher levels of sparsity.

- You examine how well the linear and nonlinear correlation coefficients identify whether two taxa are correlated [in that way]. Have you examined how well they together identify whether the association is linear or nonlinear? You comment on what you expect to see in Remark 3 (Methods section), and in your real data application, you make conclusions about it based on the pair of correlation coefficients – I’d be curious to see this in simulation results, too. That is, if you look at the pair (Pearson cor, Distance cor) how often do you see (0, 0), (0, nonzero), (nonzero, 0), and (nonzero, nonzero) in each sim setting?

5. Can you comment on how realistic the assumptions are? (The signal to noise assumption particularly caught my eye.)

Other Comments

1. Figure 1 might be easier for readers to scan quickly if the grid for 1c-e matched up with 1a-b (i.e., (T1,T1) in the upper right corner).

2. Supplementary Figure 2 – are the black points the observed data? If so, doesn't it seem like the LOESS line is a bit overfit? I don't see any reason it should swing up to a value of 5 for Ruminococcaceae with no observed values anywhere near that high. I agree that there are probably situations with nonlinear associations between two taxa – this just seems like it might not be one of those, or at least not to the degree that the LOESS curve would suggest.

3. Page 2, third from last paragraph – you state that the methods are not scale invariant when computing correlation across two or more ecosystems. Can you put this in context with an example?

- E.g., if I'm understanding correctly, it would be something like – If one wanted to compute correlation between subject A and subject B's microbiome (across all taxa), then [different total read counts?? but that doesn't matter after transforming to relative abundance... some other scaling factor?] would change your result.

4. Figure 2 - since Pearson is so dramatically bad, comparisons among the compositional methods are difficult to see. It might be worth considering whether to move the current version of the figure to the Supplement and excluding Pearson for the main text version of the figure, or vice versa. (I don't consider this vital, necessarily – just worth thinking about.)

5. Supplementary Figure 5 – you use distance correlation for the whole figure (not linear/Pearson correlation), so it seems that the legend should read something like "... for (a) linear form of association, (b) quadratic form of association".

Reviewer #3 (Remarks to the Author):

In this manuscript, Lin et al. present SECOM, a method for estimating abundance correlations between taxa in microbiome data. The method is based on modeling biases in the sampling process and finding correlations in the corrected. They use both linear and non-linear correlations. The approaches is illustrated on simulated data (where it outperforms other, widely used, methods) and on a real world dataset.

Overall, the work seems to be well done, but, at the moment, my opinion is that the contribution is limited as there is no easy reuse (#1) and there is limited

evidence that the new algorithm provides a benefit in real data (#2-3).

MAJOR COMMENTS

1. SECOM is not packaged as a tool for widespread use. The code is available in R, but it is not a package that can be easily install (for example, using bioconductor or galaxy or bioconda).

2. The authors first use simulated data where they show that their method outperforms alternative approaches. Later, they apply SECOM to a real-world dataset and show their conclusions. However, in this section (title "Illustration: Norwegian infant gut microbiome data"), they only apply their method and not any others.

3. This section on the Norwegian infant data is purely descriptive. Particularly if the authors were to add a comparison to existing methods (see #2), then it would not be a strong argument that their method is superior.

MINOR COMMENTS

4. The authors routinely use the term "linear" where they should have written "monotonical", leading to several incorrect statements, including unfairly disparaging other methods. For example: "Pearson and Spearman correlation coefficients are measures of linear associations and may not be suitable to describe nonlinear relationships." This is wrong. Spearman is very much a measure of non-linear relationships. It fails in the parabolic example ($[x-5]^2$) the authors give because that is a non-monotonic relationship.

5. The authors claim that "as demonstrated in this paper, relationships between a pair of microbes is not always linear as there exist complex interactions among them." While I believe the statement is true (and widely accepted), this paper does not demonstrate this at all! If one interprets the statement as discussing non-monotonicity (see #4 above), however, then it is not clear to me that this is as widely accepted in the field.

6. The multiple ecosystems setting could be better motivated. It is only presented for a simulated setting and it is not clear what this setting is emulating. If I understand correctly, the authors are considering for example samples from the gut and skin of the sample individuals. In particular, it seems to be different from what is provided by FlashWeave (<https://doi.org/10.1016/j.cels.2019.08.002> — a method not considered by the authors), which also provides the ability to work across ecosystems.

Response to Reviewers Comments

We thank all reviewers for their insightful and constructive comments that helped improve the content and presentation of this paper substantially. In the following we provide point by point response to the comments we received. Comments made by each reviewer are in italics and our responses follow in plain text.

Reviewer 1: *This paper SECOM proposed a sparse estimation approach to infer correlations, both linear and nonlinear, between pairs of taxa in complex microbiome community. The authors allowed for two options, with one option using Pearson/Spearman correlation which capture the linear correlation and one option using distance correlation which captures the nonlinear correlation.*

1. The model allows for the sampling fraction and sampling efficiency to differ between samples and taxa (S_i and C_j in the model), which is very appealing. These can be considered bias factors which act multiplicatively on the read counts. However, it seems that these factor only apply to the non-zero element in the count table. In other words, if a taxon has zero read count, the model assumes the zero count is accurate and S_i and C_j do not impact them. This is a limitation. You can easily imagine a situation that a taxon is in fact present in the sample, but if sequenced with low sampling efficiency, we observe zero read count for this taxon, yet if sequenced with high sampling efficiency, we observe non-zero read count. This can be problematic especially when more than 90% of the reads can be zeros. Could the authors elaborate this point, and/or assess the sensitivity of the proposed model against violations of such a simplification?

Our response: We thank the reviewer for the summary and for their important comment regarding zeros, which affects all existing methods for computing correlation coefficients, not only SECOM. This is particularly true in the case noted by the reviewer when 90% of the samples are missing for taxa. In the revision we added a Remark 5 (in Methods section) addressing this issue and we also conducted some simulation studies (Supplementary Figure 12 and Supplementary Figure 13) comparing different strategies for handling zeros. As we see from the simulation study, adding a pseudo-count to compute correlations may artificially inflate or deflate correlation coefficient, especially for rare taxa. Hence that does not solve the problem. If there are fewer missing values, then our bias corrected model may be used for imputing the missing values as commonly done in linear regression analysis. Our simulation studies indicate that correlation coefficients based on imputed data are similar to the correlation coefficients obtained using only complete case analysis, as currently implemented in SECOM. In summary, the methodology currently used in this paper appears to be reasonable, although there are opportunities to explore alternative strategies.

2. The authors provided two approaches for selecting important linear correlations, either through the “Thresholding” or “filtering” approach. However, it is not clear how the hypothesis testing (for calculating the p-values) is conducted, and how the “thresholding” is conducted.

Our response: We thank the reviewer for this comment. Basically, any method for estimating a $p \times p$ correlation matrix using a small sample size n , where $n < p$, results in an estimator that is mathematically a singular matrix although the underlying true correlation matrix is a non-singular matrix positive definite matrix. This issue applies to all methods. Consequently, the correlation matrix cannot be unbiasedly or consistently estimated by any method unless some sparsity conditions are imposed on the correlation matrix, i.e., some of the true correlation coefficients need to be constrained by zero. In

practice this is achieved by either setting up a threshold on estimated correlation coefficients or a p-value. In the two strategies we adopted, the “Thresholding” method imposes a generalized threshold on estimated correlation coefficients, and if the estimated correlation coefficient is less than the threshold then it is set to zero. We mathematically proved that the correlation matrix obtained in the “Thresholding” way is a consistent estimator (Theorem 0.3 in the Methods section). According to the p-value strategy, if the p-value is larger than a pre-specified threshold then the correlation coefficient is set to zero. For the Pearson correlation coefficient, the p-values are derived using t-distribution. For the distance correlation, the p-value is calculated using a permutation test (Szekely, G.J., Rizzo, M.L., and Bakirov, N.K. (2007)). It is important to note that the p-values are calculated not to perform a formal statistical testing but merely used as a mechanism to address sparsity. None of the approaches, whether thresholding or p-value filtering approach are designed with the intention of controlling any form of type 1 error or false discoveries. We have made this point clear in the text. We thank the reviewer for this comment.

3. How sensitive is the method to the choice of pseudo-counts, as the log transformation was applied to read counts?

Our response: Please see our response to item 1 above.

4. Is there a theoretical guarantee of FDR of the proposed method on the identified correlations? What would be the assumptions?

Our response: As noted in our response to Point 2, we are not performing any formal statistical tests but computing p-values as a mechanism for addressing sparsity. We have made this point clear in the Results section of the paper.

5. The proposed method models the read counts instead of proportions with S_i and C_j to capture the efficiency of sequencing at the sample and taxon level. Is the model consistent with a compositional model, i.e., a model that works on relative abundances, and only has one set of vectors C_j to capture the efficiency in sequencing each taxon?

Our response: We thank the reviewer for this insightful comment. Apparently, both approaches account for compositionality. In the approach suggested by the reviewer, if we move log of the library size to the right-hand side of the model, then the model proposed by the reviewer would treat the log of the library size as an off-set term in the linear model. In other words, it implicitly assumes that observed counts are proportional to the library size. On the other hand, rather than treating log of the library size as an off-set term, our model takes a more flexible approach by explicitly modeling sampling fractions, which are not only determined by library size in samples, but also microbial load in a unit volume of ecosystems of interest. The approach indicated by the reviewer will be equivalent to ours if we assume the microbial loads are the same across ecosystems.

6. The extension of the model in handling multiple ecosystems is not very clearly described.

Our response: Thanks for the comment. This comment was also made by Reviewer 2. We agree with both reviewers and hence in response to their comments, we explained for the issue of multiple ecosystems in Remark 6 (in Methods section) and illustrated SECOM and other methods using the forehead and palm data of Flores et al. (2014) in Results section. Since both forehead and palm are related to skin, it is intuitive to expect the two sites to share some common microbes that are highly

correlated. We evaluated various methods in their ability to detect these correlations. We selected the top 5 most abundant genera that are common between forehead and palm for illustration. As seen in Fig. 5a, according to SECOM, same genera from the two sites are highly correlated. Furthermore, correlations within site appear to be unchanged whether one computes correlations using the concatenated data from the two sites (Fig. 5a) or computes correlations using individual data from the two sites (Fig. 5b and Fig. 5c). However, this is not the case with other methods as seen from Supplementary Fig. 7, 8, and 9. Firstly, according to these methods none of the genera are correlated between the two skin sites, namely, forehead and palm. Furthermore, proportionality method (Supplementary Fig. 7) and SPIEC-EASI (using the MB method, Supplementary Fig. 9) found all genera to be uncorrelated (or nearly uncorrelated in the case of SPIEC-EASI) even within each site. The correlation coefficient estimates obtained by SparCC within each site changed when one computes correlations using the concatenated data from the two sites (Supplementary Fig. 8a) compared to estimates using individual data from the two sites (Supplementary Fig. 8b and Fig. 8c).

Some minor issues

1. supplemental figure 2 doesn't seem to make sense. Are both X and Y-axis log-transformed relative abundances? The number of samples are smaller than what was described in the paper. Further, there seems to have over-fitting for the LOESS curve: there was no point between 0.5 and 1, and how could the loess curve have a peak around $x = 1.5$?

Our response: A similar comment was made by Reviewer 2. We agree with both reviewers' comments on this figure. We have replaced Supplementary Fig. 2 by another figure using a data obtained from the Norwegian Microbiome study (NoMIC) study. We plotted the bias-corrected abundances of *Ruminococcaceae* and *Enterobacteriaceae* using the data obtained on day 120 after birth. While, linear fit seems reasonable (adjusted R-square = 0.53), a fourth-degree polynomial appears to fit the data better (adjusted R-square = 0.84). In more complex settings, nonlinear relationships among taxa within an ecosystem or across systems are ubiquitous (Sugihara et al., 2012).

2. There are mathematical assumptions behind the method and the theorems underlying it. It would be helpful if the readers can get some intuitive interpretation of these assumptions?

Our response: We thank the reviewer for this comment. The major assumptions made in this paper are as follows: (a) There are two major sources of bias in the observed count data, namely, sample specific sequencing efficiency and taxon specific sequencing efficiency. These sources of bias are now well accepted in the microbiome literature (McLaren et al., 2019). (b) The correlation matrix is a sparse matrix. As noted in our response to Point 2 above, this is a very common assumption that is necessary when estimating correlation and covariance matrices in high dimensions. Basically, any method for estimating a $p \times p$ correlation matrix using a small sample size n , where $n < p$, results in an estimator that is mathematically a singular matrix although the underlying true correlation matrix is a non-singular matrix positive definite matrix. This issue applies to all methods. Consequently, the correlation matrix cannot be unbiasedly or consistently estimated by any method unless some sparsity conditions are imposed on the correlation matrix, i.e., some of the true correlation coefficients need to be constrained by zero. (c) Large signal to noise ratio. This is again a very common assumption in classical linear regression analysis. If the variance of the regression model is substantially large compared to the regression coefficient, then statistical inferences on the regression parameter is under powered, unless the sample sizes are very large. Thus, in all regression analysis, to have a high power, one implicitly requires large effect sizes (signal to noise ratio). A similar phenomenon holds in the present situation.

(d) Assumption 0.3 is a mild assumption stating that empirical joint probability of observing a pair of taxa is nonzero. In other words, we avoid situations where a pair of taxa are missing in all samples, because in that case it is not possible to compute correlation coefficient between them. In the Methods section, we have added Remark 7 detailing these assumptions.

Reviewer #2: *This paper presents the method SECOM, which is a compositionally coherent approach to estimating linear (Pearson) correlation and nonlinear (distance) correlation in microbiome studies. The method is well-developed and will be useful to practitioners interested in more generic forms of association. The data application was particularly interesting - it's a very nice example of the utility of the method, as well as presenting some interesting biological conclusions. Overall, SECOM seems like a statistically valid and practically useful method.*

Our response: We thank the reviewer for the summary about our method and the paper and for their positive comments about the methodology.

Major Comments

1. *I was confused for quite a while about how the settings in the paper translate to the context of a human microbiome study -- what is an "ecosystem" in a human microbiome study: a sample? a group of samples, perhaps defined by some phenotype or disease? Are analyses within an ecosystem just analyses of samples who all belong to the same group?*

- *It would be helpful to provide some more explicit examples early in the paper and translate them into the language you use for the rest of the paper to clarify this for readers.*

Our response: We thank the reviewer for this comment. We agree that the term "ecosystem" needs to be precisely described to avoid confusion. In the introduction we now clarify various terms as suggested by the reviewer.

2. *When you discuss comparison methods, one of the critiques is that introduction of sparsity is done in a relatively arbitrary/ad hoc way. But isn't a cutoff of, say, 0.3 in SPARCC conceptually similar to your filtering method, where you choose a cutoff of alpha and set values below that to zero (although SPARCC's version is based on correlation value rather than P-value)? You also apply the Filtering approach directly to the proportionality statistic (and P-value filtering is not entirely a new idea). I'm not sure whether this is as strong a differentiator between your method and previous methods as is implied (although you do see some improvement over SPARCC's method).*

- *Relatedly, is your statement that SPARCC recommends a cutoff of 0.3 based on the sentence from their manuscript "For clarity, only edges corresponding to correlations whose magnitude is greater than 0.3 are drawn."? They also adapt the correlation estimation method to "reinforce sparsity" (their phrasing) – for accuracy, might be worth briefly mentioning that it's not a standard correlation estimate when you describe their method on page 7.*

Our response: We thank the reviewer for this comment. This comment also relates to Comment 2 of Reviewer 1. So, please see our response provided there. Basically, all methods, including SparCC and our proposed methods, recognize that when estimating a $p \times p$ correlation matrix using a small sample size n , where $n < p$, the resulting estimator is mathematically a singular matrix of a non-singular matrix. Consequently, the correlation matrix cannot be unbiasedly or consistently estimated by any method unless some sparsity conditions are imposed on the true correlation matrix. While each method for computing correlations is different, there are potential similarities in imposing sparsity constraints or thresholds. Often the thresholds used by different methods for sparsity are arbitrary. We stated in Remark 4 (in Methods section) clarifying how to properly choose the approach for sparsity in SECOM.

Specially, if one's research interest is identifying linear relationships, and the sample size is generally large (> 50 based on simulations), then the thresholding method is recommended for sparsity of the correlation matrix as it is theoretically proved to be a consistent estimator. On the contrary, if identifying general dependencies (linear and nonlinear relationships) between taxa is the primary purpose, or the sample size is limited, then the p-value filtering method is recommended for sparsity. We agree with the reviewer that although our method for estimating a correlation matrix is different from SparCC, however, conceptually the p-value filtering approach is similar to the filtering used by SparCC. Also, we thank the reviewer for pointing out the “reinforce sparsity” method for SparCC. We actually implemented the iterative SparCC in our simulation studies as well as the real data applications and have added a sentence describing it in the Methods section “More accurate estimation can be achieved by iterating the basic inference procedure. At each iteration the strongest correlated pair identified in the previous iteration is excluded, which reinforces sparsity among the remaining pairs and yields better correlation estimates.”

3. In the Discussion, it is pretty well established by now that Pearson and Spearman correlation coefficients don't work with microbiome data. In my opinion, the more interesting comparators for your method are SparCC and SPIEC-EASI, which were designed to handle microbiome data challenges and have ways to introduce sparsity. I'd prefer to see the takeaways related to these methods highlighted in the Discussion. If I understand your results correctly, the argument you'd make in that case is:

- *SECOM is better at recovering sparsity patterns than SparCC*
- *Although SECOM and SPIEC-EASI perform similarly for TPR, FPR etc. in the case of linear associations, SPIEC-EASI is more computationally intensive – and it's not great at sparsity recovery with nonlinear associations.*

(And if I haven't understood correctly, I'm doubly interested in seeing this paragraph in the Discussion!)

Our response: Again, we thank the reviewer very much for these comments. Yes, the reviewer has captured the essence of the comparisons and the methodologies correctly. As recommend by the reviewer in the Discussion section we have now highlighted these points.

4. Simulations:

- *True correlation among 50 out of 100 taxa does not seem particularly sparse to me. (Maybe your equation on page 2 is not accurately representing your sparsity pattern? That equation suggests taxon 1 is correlated with 2, 3, 4, ..., 50, and similarly for all other taxa in the first 50. The text would make me think taxon 1 was correlated with taxon 2, 2 with 1 and 3, 3 with 2 and 4, and so on.) SparCC says its sparsity assumption is strongly violated if >30% of pairs are correlated. If it's really the case that the first 50 taxa are all associated, it might be worth doing some simulations with higher levels of sparsity.*

Our response: Thanks for the comments and sorry for the confusion. In the revision of Supplementary Notes, we have clarified the simulation set-up more clearly to avoid any confusion or misunderstandings regarding the set-up.

- *You examine how well the linear and nonlinear correlation coefficients identify whether two taxa are correlated [in that way]. Have you examined how well they together identify whether the association is linear or nonlinear? You comment on what you expect to see in Remark 3 (Methods section), and in your*

real data application, you make conclusions about it based on the pair of correlation coefficients – I'd be curious to see this in simulation results, too. That is, if you look at the pair (Pearson cor, Distance cor) how often do you see (0, 0), (0, nonzero), (nonzero, 0), and (nonzero, nonzero) in each sim setting?

Our response: We thank the reviewer for the insightful comment. We have introduced a new section entitled "Concordance between SECOM (Pearson2) and SECOM (Distance) for linear and nonlinear relationships" in the Results section and added Supplementary Table 2 & 3 to address the reviewer's comments. As can be seen from our results, when taxa are linearly related and mixed with taxa that are uncorrelated, then there is a strong concordance between SECOM (Pearson2) and SECOM (Distance). Both have a very high true positive rate, and small false negative and false positive rates. Additionally, when the taxa are nonlinearly related and mixed with taxa that are uncorrelated, SECOM (Distance) has a very high true positive rate, and small false negative and false positive rates. However, SECOM (Pearson2) which is not designed for nonlinear relationships, had a small true positive rate but also had a small amount of false positive rate.

5. Can you comment on how realistic the assumptions are? (The signal to noise assumption particularly caught my eye.)

Our response: We thank the reviewer for this comment. A similar comment was made by Reviewer 1 (Point 2 in Minor comments). The major assumptions made in this paper are as follows: (a) There are two major sources of bias in the observed count data, namely, sample specific sequencing efficiency and taxon specific sequencing efficiency. These sources of bias are now well accepted in the microbiome literature (McLaren et al., 2019). (b) The correlation matrix is a sparse matrix. As noted in our response to Point 2 above, this is a very common assumption that is necessary when estimating correlation and covariance matrices in high dimensions. Basically, any method for estimating a $p \times p$ correlation matrix using a small sample size n , where $n < p$, results in an estimator that is mathematically a singular matrix although the underlying true correlation matrix is a non-singular matrix positive definite matrix. This issue applies to all methods. Consequently, the correlation matrix cannot be unbiasedly or consistently estimated by any method unless some sparsity conditions are imposed on the correlation matrix, i.e., some of the true correlation coefficients need to be constrained by zero. (c) Large signal to noise ratio. This is again a very common assumption in classical linear regression analysis. If the variance of the regression model is substantially large compared to the regression coefficient, then statistical inferences on the regression parameter is under powered, unless the sample sizes are very large. Thus, in all regression analysis, to have a high power, one implicitly requires large effect sizes (signal to noise ratio). (d) Assumption 0.3 is a mild assumption stating that empirical joint probability of observing a pair of taxa is nonzero. In other words, we avoid situations where a pair of taxa are missing in all samples, because in that case it is not possible to compute correlation coefficient between them. In the Methods section, we have added Remark 7 detailing these assumptions.

Other Comments

1. Figure 1 might be easier for readers to scan quickly if the grid for 1c-e matched up with 1a-b (i.e., (T1,T1) in the upper right corner).

Our response: Done.

2. *Supplementary Figure 2 – are the black points the observed data? If so, doesn't it seem like the LOESS line is a bit overfit? I don't see any reason it should swing up to a value of 5 for Ruminococcaceae with no observed values anywhere near that high. I agree that there are probably situations with nonlinear associations between two taxa – this just seems like it might not be one of those, or at least not to the degree that the LOESS curve would suggest.*

Our response: A similar comment was made by Reviewer 1. We agree with both reviewers' comments on this figure. We have replaced Supplementary Fig. 2 by another figure using a data obtained from the Norwegian Microbiome study (NoMIC) study. We plotted the bias-corrected abundances of *Ruminococcaceae* and *Enterobacteriaceae* using the data obtained on day 120 after birth. While, linear fit seems reasonable (adjusted R-square = 0.53), a fourth-degree polynomial appears to fit the data better (adjusted R-square = 0.84). In more complex settings, nonlinear relationships among taxa within an ecosystem or across systems are ubiquitous (Sugihara et al., 2012).

3. *Page 2, third from last paragraph – you state that the methods are not scale invariant when computing correlation across two or more ecosystems. Can you put this in context with an example?*

• *E.g., if I'm understanding correctly, it would be something like – If one wanted to compute correlation between subject A and subject B's microbiome (across all taxa), then [different total read counts?? but that doesn't matter after transforming to relative abundance... some other scaling factor?] would change your result.*

Our response: Thanks for the comment. We agree with the reviewer that the statement was not statistically precise. This comment was also made by Reviewer 1. We agree with both reviewers and hence in response to their comments, we explained for the issue of multiple ecosystems in Remark 6 (in Methods section) and illustrated SECOM and other methods using the forehead and palm data of Flores et al. (2014) in Results section. Since both forehead and palm are related to skin, it is intuitive to expect the two sites to share some common microbes that are highly correlated. We evaluated various methods in their ability to detect these correlations. We selected the top 5 most abundant genera that are common between forehead and palm for illustration. As seen in Fig. 5a, according to SECOM, same genera from the two sites are highly correlated. Furthermore, correlations within site appear to be unchanged whether one computes correlations using the concatenated data from the two sites (Fig. 5a) or computes correlations using individual data from the two sites (Fig. 5b and Fig. 5c). However, this is not the case with other methods as seen from Supplementary Fig. 7, 8, and 9. Firstly, according to these methods none of the genera are correlated between the two skin sites, namely, forehead and palm. Furthermore, proportionality method (Supplementary Fig. 7) and SPIEC-EASI (using the MB method, Supplementary Fig. 9) found all genera to be uncorrelated (or nearly uncorrelated in the case of SPIEC-EASI) even within each site. The correlation coefficient estimates obtained by SparCC within each site changed when one computes correlations using the concatenated data from the two sites (Supplementary Fig. 8a) compared to estimates using individual data from the two sites (Supplementary Fig. 8b and Fig. 8c).

4. *Figure 2 - since Pearson is so dramatically bad, comparisons among the compositional methods are difficult to see. It might be worth considering whether to move the current version of the figure to the*

Supplement and excluding Pearson for the main text version of the figure, or vice versa. (I don't consider this vital, necessarily – just worth thinking about.)

Our response: We agree with the reviewer and the standard Pearson correlation coefficient was removed in figures of the main text (Fig. 2, 3, and 4). Full comparisons including the standard Pearson correlation coefficient were provided in Supplementary Fig. 4, 5, 6.

5. Supplementary Figure 5 – you use distance correlation for the whole figure (not linear/Pearson correlation), so it seems that the legend should read something like “... for (a) linear form of association, (b) quadratic form of association”.

Our response: We thank the reviewer for the comment and have made changes accordingly.

Reviewer #3: *In this manuscript, Lin et al. present SECOM, a method for estimating abundance correlations between taxa in microbiome data. The method is based on modeling biases in the sampling process and finding correlations in the corrected. They use both linear and non-linear correlations. The approaches is illustrated on simulated data (where it outperforms other, widely used, methods) and on a real world dataset.*

Overall, the work seems to be well done, but, at the moment, my opinion is that the contribution is limited as there is no easy reuse (#1) and there is limited evidence that the new algorithm provides a benefit in real data (#2-3).

Our response: We thank the reviewer for their comments and in the following we provide point by point responses to the comments.

MAJOR COMMENTS

1. SECOM is not packaged as a tool for widespread use. The code is available in R, but it is not a package that can be easily install (for example, using bioconductor or galaxy or bioconda).

Our response: Similar to our earlier methods such as ANCOM and ANCOM-BC which are widely downloaded and used by researchers working on microbiome data, we are developing a user-friendly R function for SECOM which will be available in our ANCOMBC R package in the coming weeks. Developers of QIIME2 have already expressed interest in adopting this methodology into their software suite.

2. The authors first use simulated data where they show that their method outperforms alternative approaches. Later, they apply SECOM to a real-world dataset and show their conclusions. However, in this section (title "Illustration: Norwegian infant gut microbiome data"), they only apply their method and not any others.

Our response: In response to this reviewer's comments as well other reviewers, we have expanded the illustration section by implementing SparCC on these Norwegian infant gut microbiome data. Additionally, we applied all the methods to another data set that involves forehead and palm microbiome data obtained on the same subjects (Flores et al., 2014).

3. This section on the Norwegian infant data is purely descriptive. Particularly if the authors were to add a comparison to existing methods (see #2), then it would not be a strong argument that their method is superior.

Our response: As stated in our response to Point #2 above and in response to comments by other reviewers, we have expanded the illustration section by analyzing "palm" and "forehead" microbiome

data obtained by Flores et al. (2014). Since both forehead and palm are related to skin, it is intuitive to expect the two sites to share some common microbes that are highly correlated. We evaluated various methods in their ability to detect these correlations. We selected the top 5 most abundant genera that are common between forehead and palm for illustration. As seen in Fig. 5a, according to SECOM, same genera from the two sites are highly correlated. Furthermore, correlations within site appear to be unchanged whether one computes correlations using the concatenated data from the two sites (Fig. 5a) or computes correlations using individual data from the two sites (Fig. 5b and Fig. 5c). However, this is not the case with other methods as seen from Supplementary Fig. 7, 8, and 9. Firstly, according to these methods none of the genera are correlated between the two skin sites, namely, forehead and palm. Furthermore, proportionality method (Supplementary Fig. 7) and SPIEC-EASI (using the the MB method, Supplementary Fig. 9) found all genera to be uncorrelated (or nearly uncorrelated in the case of SPIEC-EASI) even within each site. The correlation coefficient estimates obtained by SparCC within each site changed when one computes correlations using the concatenated data from the two sites (Supplementary Fig. 8a) compared to estimates using individual data from the two sites (Supplementary Fig. 8b and Fig. 8c). Our findings in the Norwegian infant gut microbiome reveals interesting temporal changes in correlations of some families of microbiota as infants grow during their first year after birth when their gut flora also evolve. Our results provide interesting hypotheses for researchers to explore further.

MINOR COMMENTS

4. The authors routinely use the term "linear" where they should have written "monotonical", leading to several incorrect statements, including unfairly disparaging other methods. For example: "Pearson and Spearman correlation coefficients are measures of linear associations and may not be suitable to describe nonlinear relationships." This is wrong. Spearman is very much a measure of non-linear relationships. It fails in the parabolic example ($[x-5]^2$) the authors give because that is a non-monotonic relationship.

Our response: Thanks. The reviewer is correct regarding our choice of terminology. We have modified the text everywhere accordingly and added a remark (Remark 2 in Methods section) that explains various types of relationships, linear, monotonic, non-monotonic and nonlinear.

5. The authors claim that "as demonstrated in this paper, relationships between a pair of microbes is not always linear as there exist complex interactions among them." While I believe the statement is true (and widely accepted), this paper does not demonstrate this at all! If one interprets the statement as discussing non-monotonicity (see #4 above), however, then it is not clear to me that this is as widely accepted in the field.

Our response: Thanks for the comment. The distance correlation provides a valid measure for any kind of relationship, whether it is linear, monotonic but not linear or non-monotonic. If the Pearson (Spearman) correlation is zero, then one may infer lack of linear (monotonic) relationship, but one cannot rule out other forms of relationships such as non-monotonic relationships. The distance

correlation coefficient fills this important gap. We have replaced Supplementary Fig. 2 with another figure using a data obtained from the Norwegian Microbiome study (NoMIC) study. We plotted the bias corrected abundances of *Ruminococcaceae* and *Enterobacteriaceae* using the data obtained at day 120 after birth. While, linear fit seems reasonable (adjusted R-square = 0.53), a fourth-degree polynomial appears to fit the data better (adjusted R-square = 0.84). In more complex settings, nonlinear relationships among taxa within an ecosystem or across systems are ubiquitous (Sugihara et al., 2012). Hence the tool developed here is useful to quantify dependence between a pair of taxa.

6. The multiple ecosystems setting could be better motivated. It is only presented for a simulated setting and it is not clear what this setting is emulating. If I understand correctly, the authors are considering for example samples from the gut and skin of the sample individuals. In particular, it seems to be different from what is provided by FlashWeave (<https://doi.org/10.1016/j.cels.2019.08.002> — a method not considered by the authors), which also provides the ability to work across ecosystems.

Our response: We agree with the reviewer. This comment was also made by the other reviewers. As noted in our responses to other reviewers, we have provided more details regarding this in Remark 6 and have expanded the illustration section by analyzing “palm” and “forehead” microbiome data obtained by Flores et al. (2014). Please see our response to your Point 3 above.

REVIEWER COMMENTS

Reviewer #1 (Remarks to the Author):

The authors have addressed all my questions, and I have no further comment.

Reviewer #2 (Remarks to the Author):

Thank you for your thorough work on the revision of this manuscript. The clarifications and additions -- particularly related to the multiple ecosystems component of the model and the concordance between SECOM (Pearson) and SECOM (Distance) -- are very helpful.

I have a few additional questions/comments on this version of the paper:

1. It's still counterintuitive to me that complete case analysis should be the best basis on which to compute the correlation between two microbes' abundances. The simulation mimics a situation where the zeros are MCAR, but often (as you well know based on your paper's description of the two sources of bias!) the zeros are not MCAR -- possibly MAR if we have external information, possibly MNAR. CCA will definitely do well when zeros are MCAR. Do you expect your simulation approach is representative of more realistic (non-MCAR) zero patterns as well? Do you have reason to believe the bias-correction take care of this component of sparsity patterns?

1a. Adding a pseudocount of 1, while common, is a pretty drastic change for rare taxa - did you consider adding a smaller pseudocount, such as half of the relative abundance of the rarest taxon (or a count of 0.5)? What about a more nuanced zero-replacement method such as GBM (would still need to do something about taxa with only one non-zero count, but it might do better for the ones with >1 non-zero count)?

2. Results - It occurs to me that scaling is an important consideration for computation time. Does your method scale well with increasing p and/or n (empirically or theoretically)? If it takes 1.2 CPU hours for a dataset with $n=100$ and $d=200$, computation could become intractable even at sample sizes we see in real data regularly (e.g., n around 100 but d much larger, such as 1000).

2a. Does your code support parallel computation, if that's feasible for the method? If it does, that's worth noting somewhere as a strength! If it doesn't, that's not necessarily a problem, but it makes scaling of computation time a bit more important for the practicality/utility of the method.

3. Results - Illustration on temporal data. When you motivate the multiple ecosystems part of the approach, you list both different body sites (illustrated by Flores et al forehead and palm) and different time points as motivating examples. In your NoMIC application, as far as I can tell, you don't use the multiple ecosystems approach – you apply SECOM separately at each time point. Is my understanding correct? Why did you make this decision rather than the multiple-ecosystems approach? In what situations would you recommend that someone who is reading this paper use each approach? For example, if I'm interested in how taxa at day 30 are associated with those at day 120, I'd use the multiple ecosystems approach - does that come with any cost (e.g., computational), or would you use the multiple ecosystems approach even when you're primarily interested in the within-timepoint correlations?

4. Minor comments

> Remark 6 - "body cites" should be "body sites"

> Second to last paragraph of Introduction, "To quantify nonlinear...", the citation for reference 34 should be formatted as an inline reference, e.g., "... between a pair of variables, Szekely et al introduced the concept ... data.^{34}"

Reviewer #3 (Remarks to the Author):

I thank the authors for their improvements to the manuscript as they addressed all my previous concerns.

I have no further comments.

Minor typo on page 2: "It is well-known that microbiome abundance table may..." probably should read "It is well-known that microbiome abundance tables may..."

Response to Reviewers Comments

We thank the reviewers for their positive feedback on our revision. In the following we provide item by item responses to the comments provided by the reviewers.

Reviewer #1: *The authors have addressed all my questions, and I have no further comment.*

Our response: We are glad that we have successfully addressed all the comments made by the reviewer and that there are no further comments.

Reviewer #2: *Thank you for your thorough work on the revision of this manuscript. The clarifications and additions -- particularly related to the multiple ecosystems component of the model and the concordance between SECOM (Pearson) and SECOM (Distance) -- are very helpful.*

Our response: Thank you so much for your positive feedback. We appreciate it very much.

I have a few additional questions/comments on this version of the paper:

1. It's still counterintuitive to me that complete case analysis should be the best basis on which to compute the correlation between two microbes' abundances. The simulation mimics a situation where the zeros are MCAR, but often (as you well know based on your paper's description of the two sources of bias!) the zeros are not MCAR – possibly MAR if we have external information, possibly MNAR. CCA will definitely do well when zeros are MCAR. Do you expect your simulation approach is representative of more realistic (non-MCAR) zero patterns as well? Do you have reason to believe the bias-correction take care of this component of sparsity patterns?

Our response: The reviewer raises a very good point about non-MCAR mechanism of missingness. As suggested by the reviewer, we have conducted some additional simulation studies. It is widely accepted that there are two major sources of zero counts or missing taxa, namely, (a) structural zeros, or (b) rare taxa and low sequencing depth. If it is due to structural zeros, then CCA is valid as described, and adding a pseudo-count may result in misleading results (Supplementary Fig. 12a, b). However, if the zeros are due to rare taxa and low sequencing depth, from our simulation study we see that both CCA as well adding a pseudo-count yield similar result (Supplementary Fig. 12c, d). We have modified Supplementary Fig. 12 and Remark 5 in the Methods section to clarify this point.

1a. Adding a pseudocount of 1, while common, is a pretty drastic change for rare taxa - did you consider adding a smaller pseudocount, such as half of the relative abundance of the rarest taxon (or a count of 0.5)? What about a more nuanced zero-replacement method such as GBM (would still need to do something about taxa with only one non-zero count, but it might do better for the ones with >1 non-zero count)?

Our response: We thank the reviewer for the comment regarding using a different pseudo-count value and a different imputation method. Again, we conducted some additional simulation studies as suggested by the reviewer. We implemented the pseudo-count of 0.1 (instead of 1) and found little or no difference in performance when compared to pseudo-count of 1 (see below figure). Thus, in the case of structural zeros, there is no difference between a pseudo-count of 1 or some smaller value. As

suggested by the reviewer, we also considered imputation using the GBM methods (in addition to MICE which was described in the previous revision). We found that estimates based on CCA performed better than those obtained from both GBM and MICE imputed data (Supplementary Fig. 13)

2. Results - It occurs to me that scaling is an important consideration for computation time. Does your method scale well with increasing p and/or n (empirically or theoretically)? If it takes 1.2 CPU hours for a dataset with $n=100$ and $d=200$, computation could become intractable even at sample sizes we see in real data regularly (e.g., n around 100 but d much larger, such as 1000).

2a. Does your code support parallel computation, if that's feasible for the method? If it does, that's worth noting somewhere as a strength! If it doesn't, that's not necessarily a problem, but it makes scaling of computation time a bit more important for the practicality/utility of the method.

Our response: These are important comments. Scalability, while being correct, has been an important consideration for us. We clarified in the Results section (paragraph "Linear correlations") that 1.2 CPU hours was obtained due to 100 iterations for each $n/d/\alpha$ combination (thus, a total of 400 datasets). SECOM scales well with either increasing p or n . On average SECOM takes about 4 seconds for 100 samples and 200 taxa using 1 CPU core (Supplementary Table 1). The SECOM functions have been integrated into our ANCOMBC R package (<https://www.bioconductor.org/packages/release/bioc/html/ANCOMBC.html>), which also supports parallel computation.

3. Results - Illustration on temporal data. When you motivate the multiple ecosystems part of the approach, you list both different body sites (illustrated by Flores et al forehead and palm) and different time points as motivating examples. In your NoMIC application, as far as I can tell, you don't use the

multiple ecosystems approach – you apply SECOM separately at each time point. Is my understanding correct? Why did you make this decision rather than the multiple-ecosystems approach? In what situations would you recommend that someone who is reading this paper use each approach? For example, if I'm interested in how taxa at day 30 are associated with those at day 120, I'd use the multiple ecosystems approach - does that come with any cost (e.g., computational), or would you use the multiple ecosystems approach even when you're primarily interested in the within-timepoint correlations?

Our response: In response to the reviewer's comments, in addition to intra-time correlations for the NoMIC data we provided in the previous draft, we have now included inter-time, or pairwise temporal, correlations as well. We have modified the text and updated Figure 7 accordingly to reflect the new results. In addition to SECOM, we also computed pairwise temporal correlations using SparCC for these data (see Supplementary Figure 11). In contrast to SECOM, very few taxa were discovered to be temporally correlated by SparCC. This finding is consistent with the findings in forehead and palm data, as well as the simulations. When it comes to pairwise temporal correlations there are two sources of sparsity, namely, sparsity within time points and sparsity across time points because samples were not available on every infant at all three time points. Secondly, as noted in Flores et al. (2014), the temporal variability in measurements within a subject can be substantially large that can overwhelm correlations between pairs of taxa over time. Thus, unlike correlations within a time point, we expect more diffused correlations across time points. Interestingly, despite these challenges, SECOM methodology identified several pairs of taxa to be correlated across time points (see Figure 7). These correlations generate interesting hypotheses to investigate in the future.

4. Minor comments

> Remark 6 - "body cites" should be "body sites"

> Second to last paragraph of Introduction, "To quantify nonlinear...", the citation for reference 34 should be formatted as an inline reference, e.g., "... between a pair of variables, Szekely et al introduced the concept ... data.^{34}"

Our response: Thanks for these comments. We have addressed them accordingly.

Reviewer #3: *I thank the authors for their improvements to the manuscript as they addressed all my previous concerns.*

I have no further comments.

Minor typo on page 2: "It is well-known that microbiome abundance table may..." probably should read "It is well-known that microbiome abundance tables may..."

Our response: We are glad that we have successfully addressed all the comments made by the reviewer and that there are no further comments. We have now addressed the typo identified by the reviewer.

REVIEWERS' COMMENTS

Reviewer #2 (Remarks to the Author):

The authors have addressed all of my comments. Thank you for the additional comments and results!